# Structure and dynamics of cholesterol-mediated aquaporin-0 arrays and implications for lipid rafts

Po-Lin Chiu[1†‡], Juan D Orjuela[2,3†], Bert L de Groot[4], Camilo Aponte Santamaría[2,5*], Thomas Walz[1*§]

[1]Department of Cell Biology, Harvard Medical School, Boston, United States; [2]Max Planck Tandem Group in Computational Biophysics, Universidad de los Andes, Bogotá, Colombia; [3]Biomedical Engineering Department, Universidad de los Andes, Bogotá, Colombia; [4]Computational Biomolecular Dynamics Group, Max Planck Institute for Multidisciplinary Sciences, Göttingen, Germany; [5]Molecular Biomechanics Group, Heidelberg Institute for Theoretical Studies, Heidelberg, Germany

*For correspondence:
camilo.aponte@h-its.org (CAS);
twalz@rockefeller.edu (TW)

†These authors contributed
equally to this work

Present address: ‡School of
Molecular Sciences, Biodesign
Center for Applied Structural
Discovery, Arizona State
University, Tempe, United
States; §Laboratory of Molecular
Electron Microscopy, The
Rockefeller University, New York,
United States

Competing interest: The authors
declare that no competing
interests exist.

Reviewing Editor: Randy B
Stockbridge, University of
Michigan, United States

**Abstract** Aquaporin-0 (AQP0) tetramers form square arrays in lens membranes through a yet unknown mechanism, but lens membranes are enriched in sphingomyelin and cholesterol. Here, we determined electron crystallographic structures of AQP0 in sphingomyelin/cholesterol membranes and performed molecular dynamics (MD) simulations to establish that the observed cholesterol positions represent those seen around an isolated AQP0 tetramer and that the AQP0 tetramer largely defines the location and orientation of most of its associated cholesterol molecules. At a high concentration, cholesterol increases the hydrophobic thickness of the annular lipid shell around AQP0 tetramers, which may thus cluster to mitigate the resulting hydrophobic mismatch. Moreover, neighboring AQP0 tetramers sandwich a cholesterol deep in the center of the membrane. MD simulations show that the association of two AQP0 tetramers is necessary to maintain the deep cholesterol in its position and that the deep cholesterol increases the force required to laterally detach two AQP0 tetramers, not only due to protein–protein contacts but also due to increased lipid–protein complementarity. Since each tetramer interacts with four such 'glue' cholesterols, avidity effects may stabilize larger arrays. The principles proposed to drive AQP0 array formation could also underlie protein clustering in lipid rafts.

## eLife assessment

This manuscript aims to unravel the contribution of cholesterol to aquaporin-0 (AQP0) tetramer array formation within lens membranes. **Compelling** electron crystallography data are combined with **solid** molecular dynamics experiments to identify a specific cholesterol binding site of significance to protein clustering within lipid rafts. The **important** work advances our understanding of membrane biology and will be of broad interest to membrane transport biologists, biochemists, and structural biologists.

## Introduction

The current view of biological membranes is that proteins are densely packed and often organized into functional modules that perform specific biological functions (*Engelman, 2005*; *Nicolson, 2014*). Lipids play an important role in organizing membrane proteins, and the resulting specialized membrane domains are now commonly known as lipid microdomains or lipid rafts (*Karnovsky et al.,*

*1982*; *Lingwood and Simons, 2010*; *Simons and Ikonen, 1997*; *Simons and Toomre, 2000*; *Stier and Sackmann, 1973*; *Vereb et al., 2003*).

Lipid microdomains are enriched in cholesterol (*Barenholz, 2002*; *Korade and Kenworthy, 2008*; *Simons and Ikonen, 1997*) and characterized by a lipid phase that is distinct from the surrounding membrane (*Ahmed et al., 1997*; *Schroeder et al., 1991*). Cholesterol has a planar four-aromatic ring structure with a short isooctyl alkyl chain and a small 3-β-hydroxyl head group that can form a hydrogen bond with a polar group (*Figure 1—figure supplement 1A*). Because the aliphatic groups linked to the ring system are asymmetrically distributed, cholesterol features a smooth (or α) face and a rough (or β) face (*Figure 1—figure supplement 1A*). It makes up 40–90% of the lipids in eukaryotic cell membranes (*Liscum and Munn, 1999*; *van Meer et al., 2008*). Cholesterol can form non-covalent interactions with lipid acyl chains, preferentially with saturated ones (*Epand and Epand, 2004*; *Zheng et al., 2007*). As a result, the membrane can transition from a liquid-disordered to a liquid-ordered phase (*de Meyer et al., 2010*; *Ghysels et al., 2019*; *Marsh, 2010*; *Seelig, 1977*), which has been linked to the formation of lipid microdomains (*Quinn and Wolf, 2009*; *Silvius, 2003*).

Mammalian lens membranes contain a high percentage of sphingomyelin (SM) – a lipid consisting of a phosphocholine head group, a sphingosine amino alcohol, and a fatty acid (*Figure 1—figure supplement 1B*) – and cholesterol, lipids that are characteristic for lipid rafts (*Simons and Ikonen, 1997*; *Zelenka, 1984*). With a molar ratio of cholesterol to other phospholipids ranging from 2.02 to 2.52, the human lens core plasma membrane features the highest molar cholesterol content of all membranes found in human tissues (*Fleschner and Cenedella, 1991*; *Zelenka, 1984*). Aquaporin-0 (AQP0), a lens-specific water channel, is the most abundant membrane protein in lens membranes (over 60% of the total protein content) (*Alcalá et al., 1975*; *Bloemendal et al., 1972*), where it forms large two-dimensional (2D) arrays (*Gorin et al., 1984*; *Kistler and Bullivant, 1980*; *Zampighi et al., 1982*).

The propensity of AQP0 to form 2D crystals was used to determine its structure by electron crystallography (EC) of in vitro assembled 2D crystals grown with the lipid dimyristoyl phosphatidylcholine (DMPC) (*Gonen et al., 2005*). Later, AQP0 2D crystals were grown with a variety of different lipids (*Hite et al., 2015*; *Hite et al., 2010*). All these crystals had the same lattice constants, which were identical to those of AQP0 arrays in native lens membranes (*Buzhynskyy et al., 2007*). Thus, in vitro grown 2D crystals recapitulate the organization of AQP0 tetramers in 2D arrays in the native lens membrane. Interestingly, however, high-resolution EC studies, which also resolved the annular lipids (*Gonen et al., 2005*; *Hite et al., 2010*), showed that AQP0 tetramers in the 2D crystals are separated by a layer of lipids and form essentially no direct protein–protein interactions. As the crystal contacts are almost exclusively mediated by lipids, it is surprising that AQP0 2D crystals could be obtained with almost every lipid tested.

Molecular dynamics (MD) simulations further advanced our understanding of how lipids interact with AQP0. Initial simulations probing the interactions of AQP0 with lipids were carried out with pure DMPC and mixed DMPC:cholesterol membrane patches, predicting the existence of a cholesterol hotspot at the extracellular side of the AQP0 surface (*O'Connor and Klauda, 2011*). Later, MD simulations demonstrated that the positions of lipids around AQP0 seen in EC structures obtained from 2D crystals, in which the lipids are constrained by the protein crystal packing, are indeed representative of the localization of unconstrained lipids around individual AQP0 tetramers (*Aponte-Santamaría et al., 2012*). Molecular aspects, such as protein mobility and surface roughness, were identified to play key roles in defining the lipid positions (*Aponte-Santamaría et al., 2012*; *Briones et al., 2017*). Furthermore, a combination of coarse-grained and atomistic MD simulations revealed that annular lipids adopt similar positions around several members of the aquaporin family (*Stansfeld et al., 2013*), while mass spectrometry demonstrated the weak nature of the interaction of lipids with AQPZ (*Laganowsky et al., 2014*).

In vitro, AQP0 can be induced to form 2D crystals by choosing a very specific lipid-to-protein ratio (LPR) for reconstitution, but this in vitro approach does not explain why AQP0 tetramers form 2D arrays in native lens membranes, which contain an excess of lipids as well as other membrane proteins. Since lipids mediate the interactions in AQP0 2D arrays, it is likely that specific lipids play a role in the assembly of AQP0 2D arrays in the native lens membrane. As AQP0-containing lens membrane junctions are greatly enriched in cholesterol and sphingomyelin (*Fleschner and Cenedella, 1991*), lipids that play a key role in raft formation (*Brown and London, 2000*), these lipids were prime

candidates to drive AQP0 array formation. To gain structural insights into how cholesterol and/or sphingomyelin may induce AQP0 to form 2D crystals in vivo and potentially to obtain a more general understanding of how these lipids can establish lipid microdomains, we determined structures of AQP0 in membranes formed by mixtures of sphingomyelin and cholesterol and performed extensive equilibrium and force-probe MD simulations.

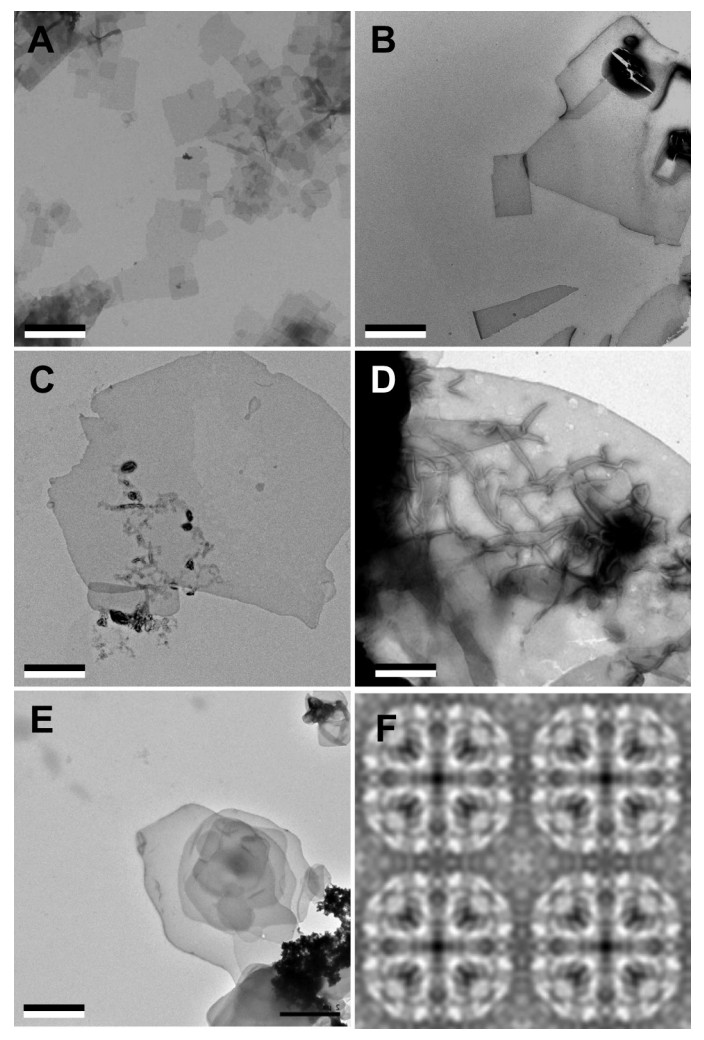

**Figure 1.** Aquaporin-0 (AQP0) forms two-dimensional (2D) crystals with all tested sphingomyelin/cholesterol mixtures. (**A–E**) AQP0 purified from sheep lenses was reconstituted with pure sphingomyelin (**A**), sphingomyelin/cholesterol mixtures at molar ratios of 2:1 (**B**), 1:2 (**C**), and 1:4 (**D**), as well as pure cholesterol (**E**). AQP0 was reconstituted under all conditions and formed diffracting 2D crystals. The scale bars are 2 µm. (**F**) Projection map of AQP0 reconstituted with pure cholesterol at 3.2 Å resolution. The 2D crystals show *p*422 symmetry and have the typical lattice constants for AQP0 crystals of *a*=*b*=65.5 Å, and *γ*=90°. The panel shows two-by-two unit cells. See also *Figure 1—figure supplements 1 and 2* and *Tables 1 and 2*.

The online version of this article includes the following figure supplement(s) for figure 1:

**Figure supplement 1.** Chemical structures of raft lipids.

**Figure supplement 2.** Diffraction of an image of an aquaporin-0 (AQP0) two-dimensional (2D) crystal grown with pure cholesterol.

**Table 1.** Internal phase residuals of all rectangular plane groups for an image of an aquaporin-0 (AQP0) crystal in a pure cholesterol membrane.

| Space group | Phase residual (degrees)* | Number of comparisons | Target residual (*degrees*)[†] |
|---|---|---|---|
| *p*1 | 12.5 [‡] | 240 | |
| *p*2 | 33.8 | 120 | 17.8 |
| *p*12b | 14.5 [§] | 108 | 12.9 |
| *p*12a | 31.7 | 109 | 12.9 |
| *p*12₁b | 83.7 | 108 | 12.9 |
| *p*12₁a | 80.0 | 109 | 12.9 |
| *c*12b | 14.5 [§] | 108 | 12.9 |
| *c*12a | 31.7 | 109 | 12.9 |
| *p*222 | 27.0 | 337 | 14.4 |
| *p*222₁b | 60.0 | 337 | 14.4 |
| *p*222₁a | 68.2 | 337 | 14.4 |
| *p*22₁2₁ | 69.7 | 337 | 14.4 |
| *c*222 | 27.0 | 337 | 14.4 |
| *p*4 | 26.3 | 352 | 14.3 |
| **p422** [¶] | **24.5** | **791** | **13.3** |
| *p*42₁2 | 62.6 | 791 | 13.3 |

Internal residuals were determined using the ALLSPACE program (**Valpuesta et al., 1994**) using spots from IQ1 to IQ5 to a resolution of 6 Å.

*Phase residual versus other spots (90° random).

[†]Target residual based on the statistics taking Friedel weight into account.

[‡]Note that no phase comparison is possible in space group p1, so that the listed numbers are theoretical phase residuals based on the signal-to-noise ratio of the observed diffraction spots in the Fourier transform.

[§]Within 5° of target residual.

[¶]The symmetry indicated in bold was used to calculate the final 2D projection map.

# Results

## AQP0 forms 2D crystals with all tested cholesterol/sphingomyelin mixtures

Purified AQP0 was reconstituted with mixtures of sphingomyelin and cholesterol at molar ratios of 1:0, 3:1, 2:1, 1:1, 1:2, 1:4, and 0:1 (*Figure 1A–E*). Every lipid mixture yielded membranes, which had a sheet-like morphology at sphingomyelin:cholesterol ratios of up to 1:2 and formed vesicles at higher cholesterol percentages. In every case, power spectra of images taken of trehalose-embedded samples showed diffraction spots consistent with typical 2D arrays of AQP0 (data not shown).

Notably, even reconstitution of AQP0 with pure cholesterol yielded vesicles, which displayed a tendency to stack (*Figure 1E*). Cholesterol on its own does not form bilayers under physiological conditions and requires a molecule of complementary shape to do so (*Kumar, 1991*; *Raguz et al., 2011*). AQP0 appears to fulfill this requirement as it does form membranes with cholesterol. However, reconstitution of AQP0 with cholesterol only yielded membranes within a narrow LPR range, from 0.2 to 0.4. At higher or lower LPRs, only protein aggregates and no membranes were observed.

Fourier transforms of cryo-EM images of trehalose-embedded AQP0 2D crystals formed with cholesterol at an LPR of 0.4 showed clear reflections (*Figure 1—figure supplement 2*), establishing that the crystals have the same lattice constants as all other AQP0 2D crystals analyzed to date (*a*=65.5 Å, *b*=65.5 Å, and *γ*=90°). Phase comparisons of the reflections showed that the crystals have *p*422 plane symmetry (*Table 1*) and thus have to be double-layered. Merging of 15 images yielded a projection map at 3.2 Å resolution (*Figure 1F* and *Table 2*). Although the crystals were not of sufficient

**Table 2.** Phase residuals for the merging of 15 images of aquaporin-0 (AQP0) two-dimensional (2D) crystals in pure cholesterol membranes.

| Plane group symmetry | p422 |
| --- | --- |
| Unit cell dimensions | a=65.5 Å, b=65.5 Å, and γ=90° |
| Number of processed electron micrographs | 15 |
| Resolution limit for merging | 3.0 Å |
| Number of phases | 2773 |
| Phase residuals in resolution bins | |
| 1000.0 Å – 11.6 Å | 21.7° |
| 11.6 Å – 8.2 Å | 31.2° |
| 8.2 Å – 6.7 Å | 34.1° |
| 6.7 Å – 5.8 Å | 47.2° |
| 5.8 Å – 5.2 Å | 43.7° |
| 5.2 Å – 4.7 Å | 50.4° |
| 4.7 Å – 4.4 Å | 58.0° |
| 4.4 Å – 4.1 Å | 62.3° |
| 4.1 Å – 3.9 Å | 62.3° |
| 3.9 Å – 3.7 Å | 64.0° |
| 3.7 Å – 3.5 Å | 70.8° |
| 3.5 Å – 3.4 Å | 69.1° |
| 3.4 Å – 3.2 Å | 71.0° |
| 3.2 Å – 3.1 Å | 88.0° |
| 3.1 Å – 3.0 Å | 86.0° |
| Overall | 54.5° |

quality to determine a high-resolution three-dimensional (3D) density map, the 2D projection map shows that AQP0 tetramers in a pure cholesterol membrane are organized in the same way as in membranes formed with all other lipids analyzed to date.

### Structure determination of AQP0$_{2SM:1Chol}$

Fourier transforms of images of trehalose-embedded AQP0 crystals obtained with sphingomyelin/cholesterol mixtures showed reflections to a resolution of about 2 Å (*Chiu et al., 2015*). We focused first on AQP0 2D crystals obtained with a sphingomyelin/cholesterol mixture at a molar ratio of 2:1, from here on referred to as AQP0$_{2SM:1Chol}$. We collected electron diffraction patterns of these crystals at different tilt angles under low-dose conditions. While diffraction patterns recorded from untilted crystals showed reflections to a resolution of 2 Å (*Figure 2A*), reflections in diffraction patterns from highly tilted crystals were not visible beyond a resolution of 2.5 Å (*Figure 2—figure supplement 1A*).

After merging 214 diffraction patterns from crystals tilted up to ~72° and phasing the intensity dataset by molecular replacement, we were able to calculate a 3D density map at 2.5 Å resolution (*Figure 2B*), which made it possible to model the AQP0 structure. To build the sphingomyelin molecules, we initially only modeled the head groups and 10 carbon atoms for each of the two acyl chains. If the 2$F_o$-$F_c$ map after crystallographic refinement showed additional density, we extended the acyl chains, and this cycle was iterated. To avoid over-fitting, after each iteration, we assessed the values for $R_{work}$, $R_{free}$, and their difference, as well as the consistency of the calculated 2$F_o$-$F_c$ map with the composite omit map. The final model includes seven sphingomyelin molecules, SM1–SM7, with acyl chains ranging from 11 to 16 carbon atoms in length (*Figure 3A*). The one density in the map that was consistent with the characteristic four-ring structure of cholesterol was modeled as Chol1 (*Figure 3A*).

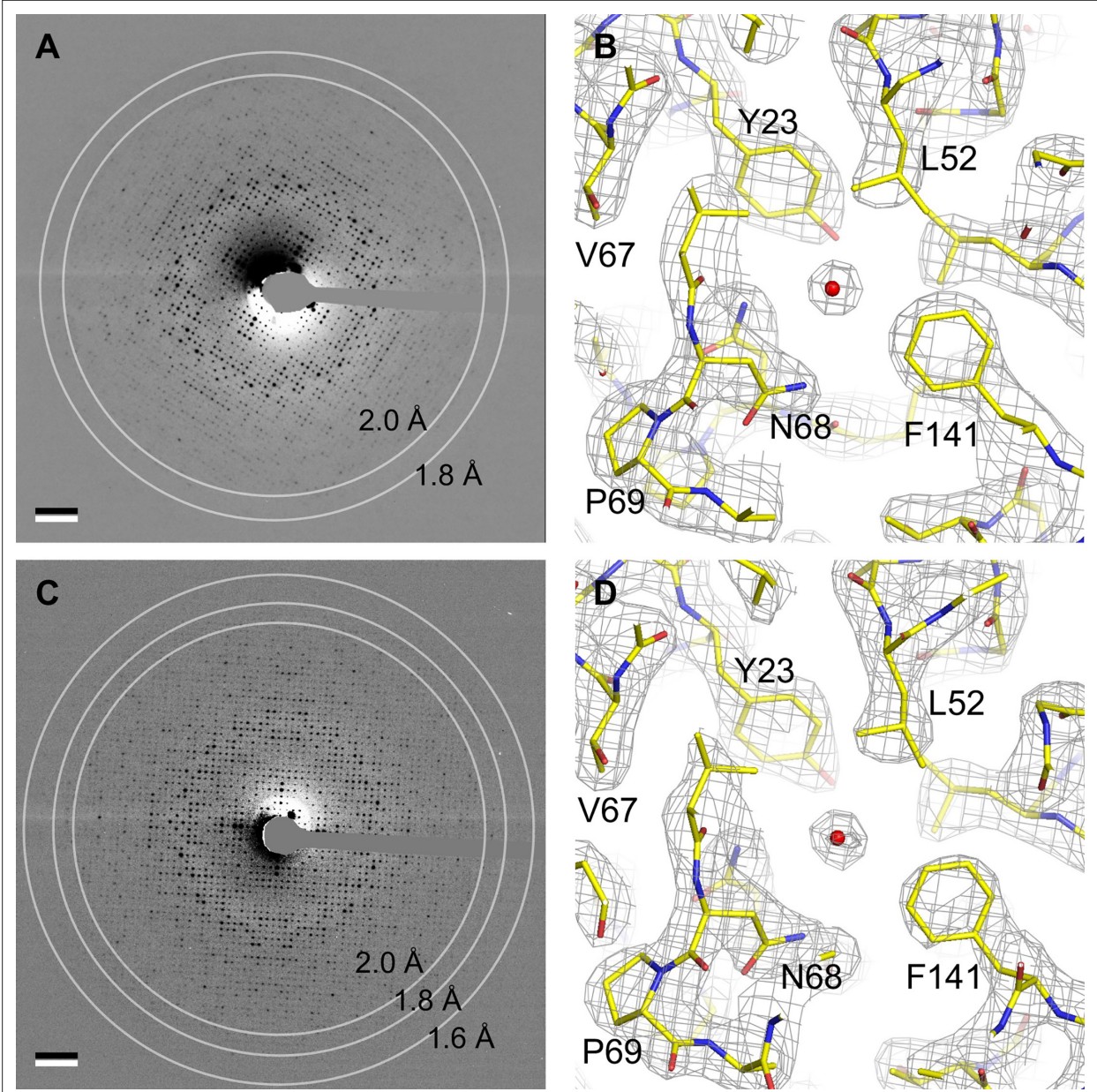

**Figure 2.** Electron crystallography provides structures of aquaporin-0 (AQP0) in sphingomyelin/ cholesterol bilayers at 2.5 Å resolution. (**A**) Electron diffraction pattern of an untilted AQP0 two-dimensional (2D) crystal reconstituted at a sphingomyelin:cholesterol ratio of 2:1, showing reflections to ~2 Å resolution. Scale bar indicates (10 Å)$^{-1}$. (**B**) Density map at 2.5 Å resolution used to build the AQP0$_{2SM:1Chol}$ structure. A region of the water-conducting pathway close to the NPA (asparagine-proline-alanine), the AQP signature motif, is shown. The 2$F_o$-$F_c$ map contoured at 1.5$\sigma$ is shown as gray mesh, the AQP0 model is shown in yellow with oxygen atoms in red and nitrogen atoms in blue. The red sphere represents a water molecule. (**C**) A diffraction pattern of an untilted AQP0 2D crystal reconstituted at a sphingomyelin:cholesterol ratio of 1:2, showing reflections to better than 1.6 Å resolution. Scale bar indicates (10 Å)$^{-1}$. (**D**) Density map at 2.5 Å resolution used to build the AQP0$_{1SM:2Chol}$ structure. The same region as in (**B**) is shown with the same color code. See also *Figure 2—figure supplement 1*.

The online version of this article includes the following figure supplement(s) for figure 2:

**Figure supplement 1.** Electron diffraction patterns of two-dimensional (2D) crystals reconstituted with sphingomyelin/cholesterol mixtures tilted to 60°.

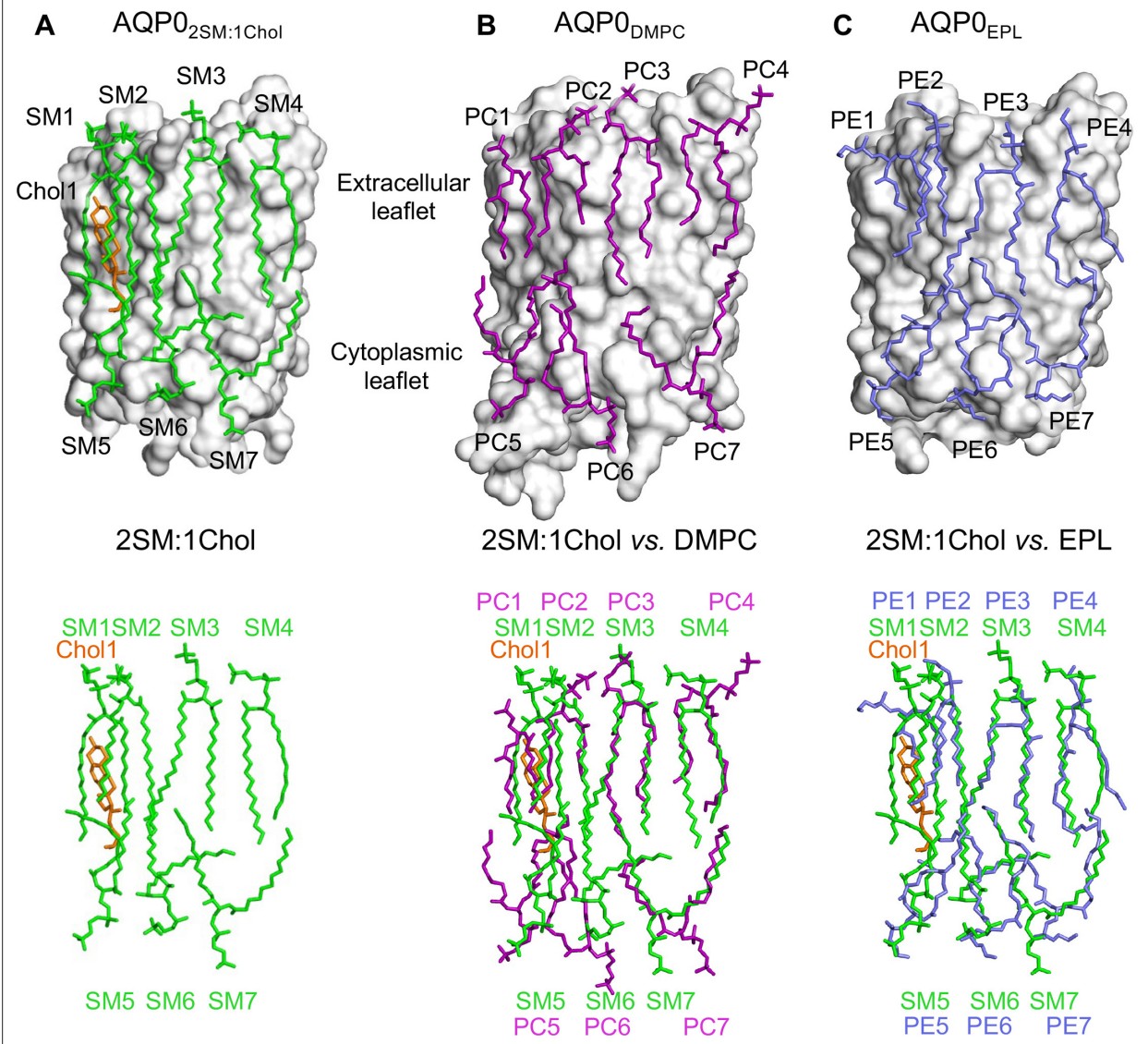

**Figure 3.** The 2:1 sphingomyelin/cholesterol bilayer surrounding aquaporin-0 (AQP0) is similar to bilayers formed by phosphoglycerolipids. (**A**) The top panel shows the seven sphingomyelins (light green sticks) and one cholesterol (orange sticks) molecules forming the bilayer around an AQP0 subunit (gray surface). The bottom panel shows just the lipid bilayer. (**B**) The top panel shows dimyristoyl phosphatidylcholine (DMPC) lipids (purple sticks) surrounding an AQP0 subunit (*Gonen et al., 2005*) and the bottom layer shows an overlay of the DMPC bilayer with the 2:1 sphingomyelin/cholesterol bilayer. (**C**) The top panel shows an *Escherichia coli* polar lipids extract (EPL) bilayer (modeled as PE lipids) (light brown sticks) surrounding an AQP0 subunit (*Hite et al., 2010*) and the bottom layer shows an overlay of the EPL bilayer with the 2:1 sphingomyelin/cholesterol bilayer. See also *Figure 3—figure supplements 1–3* and *Table 3*.

The online version of this article includes the following figure supplement(s) for figure 3:

**Figure supplement 1.** Distribution of sphingomyelin and cholesterol molecules around aquaporin-0 (AQP0) in two-dimensional (2D) crystals.

**Figure supplement 2.** Structure comparisons of aquaporin-0 (AQP0) in different lipid bilayers.

**Figure supplement 3.** Interactions of sphingomyelin lipids in AQP0$_{2SM:1Chol}$ with aquaporin-0 (AQP0) and Chol1.

**Table 3.** Statistics of electron crystallographic structures for aquaporin-0 (AQP0) in sphingomyelin/cholesterol membranes.

| | 2SM:1Chol | 1SM:2Chol |
|---|---|---|
| *Crystal parameters* | | |
| Space group | *p*422 | |
| Unit cell constants | *a*=*b*=65.5 Å, *γ*=90° | |
| Assumed thickness | 200 Å | |
| *Electron diffraction* | | |
| Diffraction patterns | 214 | 241 |
| | (0°:16; 20°:19; 45°:77; 60°:84; 65°:17; 70°:1) | (0°:15; 20°:18; 45°:51; 60°:86; 65°:30; 70°:41) |
| Maximum tilt angle | 71.72° | 72.35° |
| Upper resolution limit for merging | 2.3 Å | 2.3 Å |
| $R_{Friedel}$ | 0.149 | 0.138 |
| $R_{merge}$ | 0.216 | 0.199 |
| Observed amplitudes | 122,501 | 127,703 |
| Unique reflections | 16,437 | 17031 |
| Minimum of $F_{obs}$/$Sigma_{obs}$ | 1.33 | 1.33 |
| Fourier space sampled | 87.0% | 90.7% |
| Multiplicity | 6.2 | 6.3 |
| | (2.3 Å: 4.9) | (2.3 Å: 4.6) |
| *Crystallographic refinement* | | |
| Resolution range | 13.8–2.35 Å | 11.8–2.35 Å |
| $R_{work}$/$R_{free}$* | 0.260/0.286 | 0.262/0.287 |
| Atoms | | |
| Protein | 1663 | 1663 |
| Lipids | 336 | 341 |
| Water | 11 | 11 |
| RMS deviation | | |
| Bond length (Å) | 0.008 | 0.005 |
| Bond angle (degrees) | 1.096 | 0.842 |
| *Model validation* | | |
| Clashscore | 8.86 | 6.98 |
| MolProbity | 1.79 | 1.79 |
| Rotamer outliers (%) | 0.00 | 0.60 |
| C-beta deviation | 0 | 0 |
| Ramachandran plot (%) | | |
| Disallowed | 0.00 | 0.00 |
| Allowed | 4.59 | 5.96 |
| Favored | 95.41 | 94.04 |

*$R_{free}$ was calculated from randomly selected 10% of total reflections, and $R_{work}$ was calculated from the remaining 90% of reflections.

The density in the center of four adjacent AQP0 tetramers around the fourfold symmetry axis was poorly resolved and could not be modeled (*Figure 3—figure supplement 1B*). The statistics for EC data collection, model building, refinement, and validation of AQP0$_{2SM:1Chol}$ are summarized in *Table 3*.

### The structure of AQP0$_{2SM:1Chol}$ shows an AQP0 array with a low cholesterol content

AQP0 in the 2:1 sphingomyelin/cholesterol membrane adopts the same conformation as it does in a DMPC bilayer, AQP0$_{DMPC}$ (*Gonen et al., 2005*), and in an *E. coli* polar lipid extract (EPL) bilayer, AQP0$_{EPL}$ (*Hite et al., 2010*), with root mean square deviation (RMSD) between the Cα atoms of 0.271 Å and 0.264 Å, respectively (*Figure 3—figure supplement 2A and B*). The densities of the side chains along the water permeation pathway were clearly resolved as were the water molecules (*Figure 2B*). The water pathway through the AQP0 subunits showed density for four water molecules, at positions similar to those seen in previous EC AQP0 structures (*Gonen et al., 2004*; *Hite et al., 2010*).

Comparison of the lipid bilayer in the AQP0$_{2SM:1Chol}$ structure (*Figure 3A*) with those in the AQP0$_{DMPC}$ and AQP0$_{EPL}$ structures (*Figure 3B and C*) shows that the seven sphingomyelin molecules assume similar positions as the DMPC and EPL lipids, further strengthening the earlier conclusion that annular lipids are not randomly distributed around a membrane protein but assume preferred positions (*Hite et al., 2010*). The density for sphingomyelin SM3 is well defined. The carbonyl oxygen of its head group is hydrogen-bonded with the tyrosyl side chain of Tyr105 and the amide oxygen of the guanidinium side chain of Arg196 (*Figure 3—figure supplement 3A*). However, the head groups of the remaining sphingomyelin lipids do not form specific interactions with AQP0, as reported for the DMPC and EPL head groups in the previous AQP0$_{DMPC}$ and AQP0$_{EPL}$ structures. Also, as observed in those structures, the sphingomyelin acyl chains follow grooves on the surface of AQP0.

The only cholesterol molecule, Chol1, resolved in the AQP0$_{2SM:1Chol}$ structure is located in the extracellular leaflet (*Figure 3A*). The tetracyclic ring of Chol1 makes π-stacking interactions with AQP0 residues His201 and Trp205, and its alkyl tail makes van der Waals interactions with residues Ile87 and Val90 (*Figure 3—figure supplement 3B*). Chol1 also interacts extensively with the adjacent sphingolipid in the extracellular leaflet, SM2, the acyl chains of which assume an all *anti*-dihedral conformation, increasing their conformational order (*Figure 3—figure supplement 3B*).

While Chol1 may increase the conformational order of adjacent SM2, this effect is unlikely to induce AQP0 to form an array. Similarly, the location of Chol1 with respect to adjacent AQP0 tetramers (*Figure 3—figure supplement 1B*) does not provide any clues as to how it could affect array formation in situ. We therefore decided to analyze the structure of AQP0 in a membrane that has a higher cholesterol content.

### Structure determination of AQP0$_{1SM:2Chol}$

Lens membranes contain more cholesterol than is represented by the AQP0$_{2SM:1Chol}$ structure. Therefore, we determined the structure of AQP0 2D crystals that were grown at a molar sphingomyelin:cholesterol ratio of 1:2, which is close to the lipid composition of human lens core membranes (*Fleschner and Cenedella, 1991*; *Zelenka, 1984*). These crystals were even better ordered, and electron diffraction patterns of untilted specimens showed reflections to a resolution better than 1.6 Å (*Figure 2C*). Diffraction patterns of 2D crystals tilted to 60° showed reflections to a resolution of ~2.5 Å (*Figure 2—figure supplement 1B*). We used the same data collection and processing scheme for AQP0$_{2SM:1Chol}$ to obtain a density map for AQP0$_{1SM:2Chol}$ at 2.5 Å resolution (*Figure 2D*). The density map allowed us to model AQP0, five sphingomyelin molecules, SM2, SM3, SM5, SM6, and SM7 (numbers corresponding to the sphingomyelin molecules in AQP0$_{2SM:1Chol}$), as well as four cholesterols (*Figure 4A*). As with the AQP0$_{1SM:2Chol}$ structure, it was not possible to model lipids in the area near the fourfold axis, where four AQP0 tetramers come together (*Figure 3—figure supplement 1D*). The statistics for the AQP0$_{1SM:2Chol}$ structure are summarized in *Table 3*.

### The structure of AQP0$_{1SM:2Chol}$ shows an AQP0 array with a high cholesterol content

Superimposition of the AQP0$_{1SM:2Chol}$ structure with the AQP0$_{2SM:1Chol}$, AQP0$_{DMPC}$, and AQP0$_{EPL}$ structures yielded RMSD values between the Cα atoms of 0.431 Å, 0.390 Å, and 0.362 Å (*Figure 3—figure supplement 2C–E*), again showing that the lipid environment has no detectable effect on the

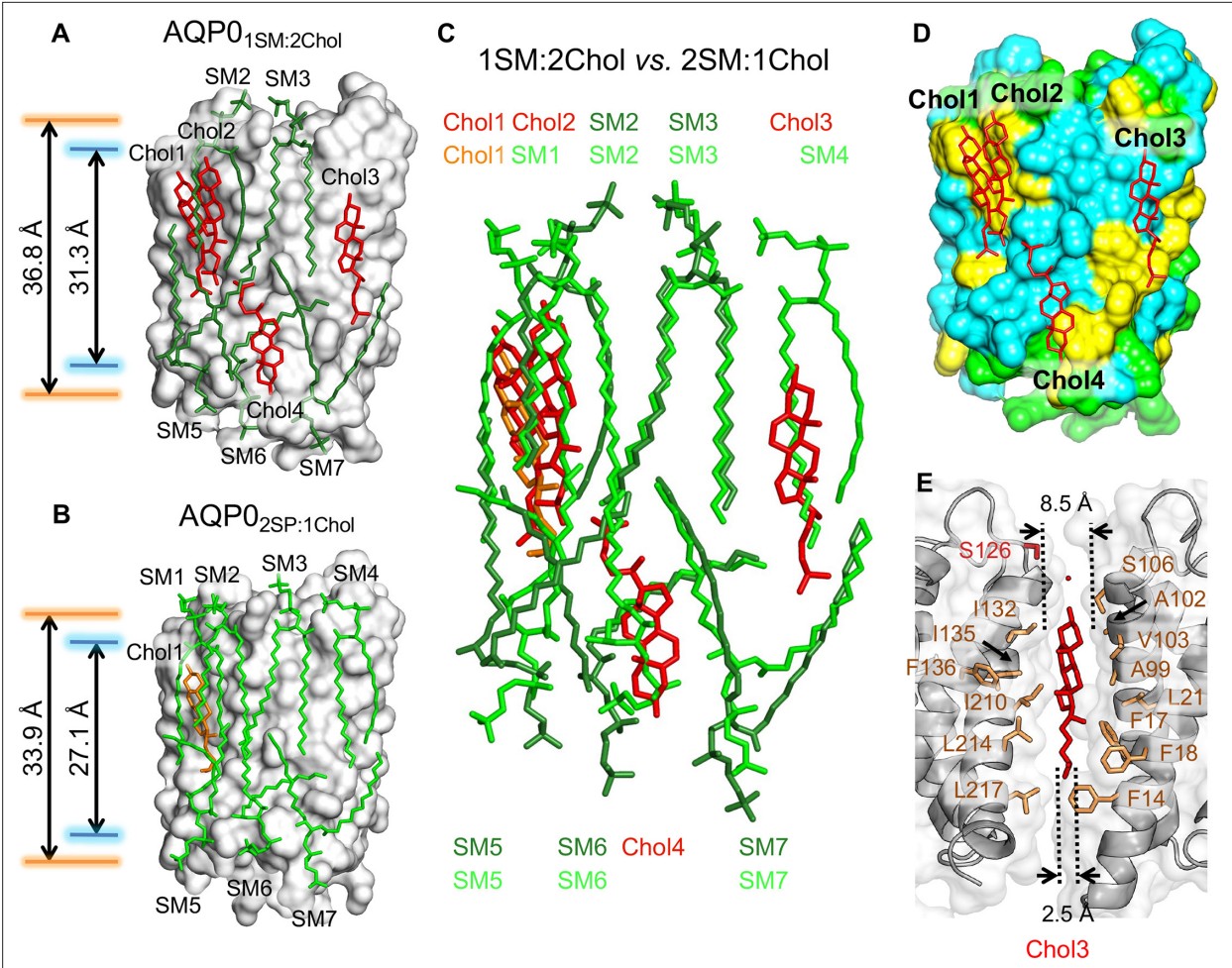

**Figure 4.** The 1:2 sphingomyelin/cholesterol bilayer surrounding aquaporin-0 (AQP0) and comparison with the 2:1 sphingomyelin/cholesterol bilayer. (**A**) The five sphingomyelins (dark green sticks) and four cholesterol (red sticks) molecules surrounding an AQP0 subunit (gray surface). The arrows between the orange and blue lines indicate the average distances between the phosphorus atoms of the phosphodiester groups and the nitrogen atoms of the amide groups in the two leaflets, respectively. (**B**) The AQP0$_{2SM:1Chol}$ structure shown for comparison with the AQP0$_{1SM:2Chol}$ structure in (**A**). Arrows as in (**A**). (**C**) Overlay of the lipid bilayers in the AQP0$_{2SM:1Chol}$ and AQP0$_{1SM:2Chol}$ structures. (**D**) Location of the four cholesterols (red sticks) in the AQP0$_{1SM:2Chol}$ structure with respect to AQP0 surface characteristics. Color coding: yellow, aromatic residues; cyan, hydrophobic residues; and light green, polar and charged residues. (**E**) Position of cholesterol Chol3 (red sticks) in the AQP0$_{1SM:2Chol}$ structure and its interaction with residues of two adjacent AQP0 tetramers (brown sticks). The dotted lines indicate the distance between the two adjacent AQP0 tetramers at the positions of the ring system (~8.5 Å) and the acyl chain (~2.5 Å). See also *Figure 4—figure supplements 1 and 2* and *Table 3*.

The online version of this article includes the following figure supplement(s) for figure 4:

**Figure supplement 1.** Interactions of cholesterol molecules in AQP0$_{1SM:2Chol}$ with aquaporin-0 (AQP0).

**Figure supplement 2.** Average *B*-factors of acyl chains in structures of aquaporin-0 (AQP0) in different lipid bilayers.

**Figure supplement 3.** The hydroxyl head group of Chol3 makes a hydrogen bond with a water molecule.

conformation of AQP0. Also, the water pathway in AQP0$_{1SM:2Chol}$ showed the same densities representing water molecules as observed in the AQP0$_{2SM:1Chol}$ structure. Hence, the AQP0 structure is essentially identical in the two analyzed sphingomyelin/cholesterol bilayers.

In the AQP0$_{2SM:1Chol}$ structure, an AQP0 subunit is surrounded by seven sphingomyelin molecules, but in the AQP0$_{1SM:2Chol}$ structure, due to the higher cholesterol content, two of the sphingomyelins in the extracellular leaflet, SM1 and SM4, have been replaced by cholesterols (*Figure 4A and B*). Of the two sphingomyelins remaining in the extracellular leaflet, the conformation of SM3 is virtually identical to that in the AQP0$_{2SM:1Chol}$ structure (*Figure 4C*). The other sphingomyelin in the extracellular leaflet, SM2, as well as the three sphingomyelins in the cytoplasmic leaflets, SM5–SM7, all occupy similar positions as their counterparts in the AQP0$_{2SM:1Chol}$ structure, but adopt different conformations. In particular, SM2 and

SM6 adapt their conformation to accommodate the additional cholesterol molecules. With the exception of SM3, the head group of which is stabilized by identical interactions with AQP0 in AQP0$_{2SM:1Chol}$ and AQP0$_{1SM:2Chol}$, the head groups of all other corresponding SM are different in the two structures (*Figure 4C*), corroborating the general lack of specific lipid–protein interactions in the head-group region.

The four cholesterol molecules in the AQP0$_{1SM:2Chol}$ structure show an interesting distribution. Overlaying the two bilayers shows that Chol1 in AQP0$_{1SM:2Chol}$ almost perfectly overlaps with Chol1 in AQP0$_{2SM:1Chol}$ (*Figure 4C*), and Chol1 interacts with AQP0 in the same way in the two structures, strengthening the notion that this is the preferred location for cholesterol to interact with AQP0. In the AQP0$_{2SM:1Chol}$ structure, Chol1 also interacts extensively with the adjacent sphingolipids, but some of these interactions are absent in the AQP0$_{1SM:2Chol}$ structure, due to the presence of the additional Chol2 (*Figure 4A*). The interaction with extracellular sphingomyelin SM1 in AQP0$_{2SM:1Chol}$ is replaced by an interaction with an acyl chain from cytoplasmic sphingomyelin SM5 (*Figure 4C*). Notably, however, despite their proximity, there do not seem to be any direct interactions between the two cholesterols (*Figure 4—figure supplement 1A*).

In addition to Chol1 and Chol2 in the extracellular leaflet, another cholesterol, Chol4, is located in the cytoplasmic leaflet (*Figure 4A*). Like Chol1 and Chol2, Chol4 interacts with AQP0 surface areas that feature aromatic residues (*Figure 4D*), in particular tryptophan and phenylalanine residues, which appear to make π-stacking interactions with the cholesterol ring system (*Burley and Petsko, 1985*; *McGaughey et al., 1998*; *Figure 4—figure supplement 1*).

Chol3 is the most unusual cholesterol molecule seen in the AQP0$_{1SM:2Chol}$ structure. While all other cholesterols are either within the extracellular or cytoplasmic leaflet, Chol3 is located almost in the middle of the bilayer, with its hydroxyl head group located in the middle of the extracellular leaflet (*Figure 4—figure supplement 3*). It sits in a pocket between the two adjacent AQP0 tetramers that is wider in the extracellular leaflet than the cytoplasmic leaflet (*Figure 4E*). The orientation of Chol3 is that of the other cholesterols in the extracellular leaflet, suggesting that it originated from that leaflet. There are three phenylalanine residues in the vicinity of Chol3, but these do not form π-stacking interactions with its ring system but are close to its acyl chain (*Figure 4E*).

To assess the influence of cholesterol content on the hydrophobic thickness of the membrane, we measured the average distance between the phosphorus atoms of the phosphodiester groups and the nitrogen atoms of the amide groups of the SM in the two leaflets (*Figure 1—figure supplement 1*), which were 36.8 Å and 31.3 Å, respectively, for the AQP01SM:2Chol bilayer (*Figure 4A*) and 33.9 Å and 27.1 Å, respectively, for the AQP02SM:1Chol bilayer (*Figure 4B*).

## Cholesterol positions observed in the EC structures are representative of those around single AQP0 tetramers

The EC structures of AQP0 were obtained with 2D crystals, in which the lipids are constrained by the packing of the AQP0 tetramers. Therefore, we performed MD simulations and calculated time-averaged densities to investigate if the cholesterol positions seen in these structures represent the positions unconstrained cholesterol would adopt around individual AQP0 tetramers, as has previously been shown to be the case for DMPC lipids (*Aponte-Santamaría et al., 2012*). The EC AQP0 structures obtained with SM:Chol ratios of 2:1, AQP0$_{2SM:1Chol}$, and 1:2, AQP0$_{1SM:2Chol}$, revealed distinct localization patterns for cholesterol around AQP0. In the AQP0$_{2SM:1Chol}$ structure, a single cholesterol molecule was observed, associated with AQP0 surface S1 (*Figure 5*, top). This cholesterol position was also identified in a previous MD simulation study (*O'Connor and Klauda, 2011*). In the density map representing the average localization of cholesterol around AQP0 over time that we calculated from MD simulations of an individual AQP0 tetramer in a 2:1 SM:Chol membrane (*Figure 5—figure supplement 1B*), we also found a high cholesterol density in the same position (*Figure 5*, top). The simulations identified an additional hotspot, located on the extracellular side of surface S1 (see high density marked with an asterisk in *Figure 5*, top). No cholesterol was observed in this position in the AQP0$_{2SM:1Chol}$ structure, suggesting that it may be preferentially occupied by cholesterol around single AQP0 tetramers but may be unfavorable in the context of AQP0 arrays. In contrast, the AQP0$_{1SM:2Chol}$ structure revealed four cholesterol molecules associated with the AQP0 surface (*Figure 5*, bottom), including the one seen at the SM:Chol ratio of 2:1. Our simulations at that high cholesterol concentration largely recapitulated these positions, as seen by the high cholesterol densities at positions for Chol1, Chol2, Chol3, and Chol4 (*Figure 5*, bottom).

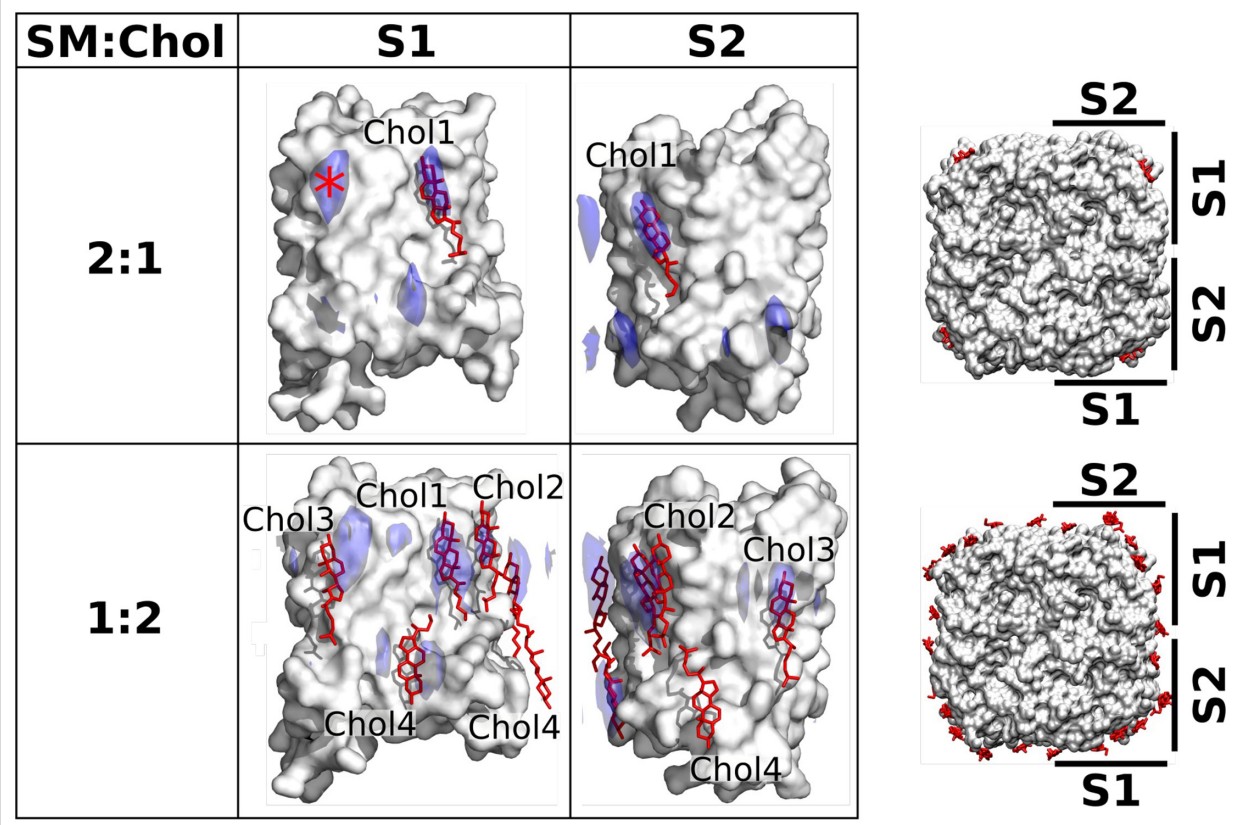

**Figure 5.** Localization of cholesterol around aquaporin-0 (AQP0) monomers from unbiased molecular dynamics (MD) simulations of individual AQP0 tetramers in sphingomyelin (SM) membranes with low and high cholesterol concentration. Density maps representing the localization of cholesterol around AQP0 over time were computed from simulations starting from unbiased cholesterol positions in membranes at the indicated SM:Chol ratios. After combining the four maps calculated individually for the four subunits of the tetramer, cholesterol densities were projected (blue areas) onto the surface of a single AQP0 monomer (white surface). Projections are shown for the S1 and S2 monomer surfaces, as defined in the representations to the right. Lipids seen in the electron crystallographic structures obtained in membranes at the respective SM:Chol ratios are displayed as sticks and labeled according to the electron crystallographic structures. Densities are contoured at $10\sigma$ for the 2:1 SM:Chol membrane and at $9\sigma$ for the 1:2 SM:Chol membrane. The density hotspot indicated with an asterisk coincides with the Chol3 position seen at the 1:2 SM:Chol ratio. See also *Figure 5—figure supplements 1–3*.

The online version of this article includes the following figure supplement(s) for figure 5:

**Figure supplement 1.** Molecular dynamics simulations of aquaporin-0 (AQP0) in sphingomyelin (SM)/cholesterol (Chol) membrane systems.

**Figure supplement 2.** Deuterium order parameters for the membrane patches in equilibration.

**Figure supplement 3.** Average area-per-lipid (APL) for the membrane patches in equilibration.

A particularly interesting cholesterol in the AQP0$_{1SM:2Chol}$ structure is Chol3, which is localized close to the center of the lipid bilayer and is therefore referred to as 'deep cholesterol'. Our density map representing the average cholesterol localization over time calculated for the high cholesterol concentration shows high density close to the position of Chol3, on both surfaces S1 and S2, but this density is localized to the extracellular leaflet rather than the middle of the membrane (*Figure 5*). This finding suggests that the deep cholesterol is not stable when cholesterol associates with individual AQP0 tetramers.

Thus, our simulations demonstrate, as previously shown for DMPC lipids (*Aponte-Santamaría et al., 2012*), that the crystalline packing of AQP0 tetramers in 2D crystals does, in general, not affect the positions where cholesterol preferentially interacts with AQP0. However, notably, deep-binding Chol3 seen in the AQP0$_{1SM:2Chol}$ structure is not represented in the MD simulations using an isolated AQP0 tetramer. Therefore, we performed further simulations with pairs of AQP0 tetramers to assess whether the deep position of Chol3 represents a crystallographic artifact or may be the result of interface properties between two neighboring AQP0 tetramers.

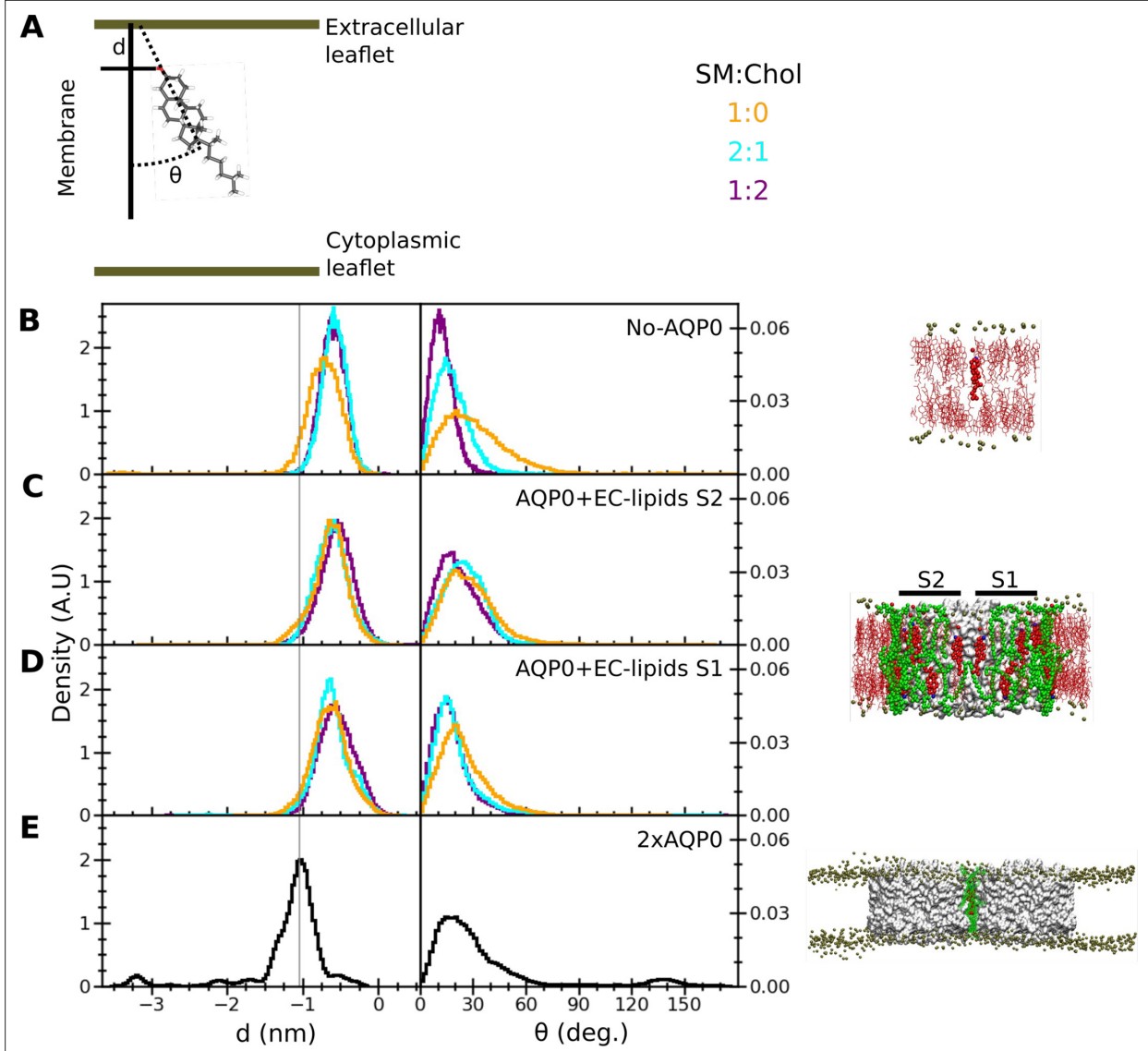

**Figure 6.** Insertion depth and orientation of a cholesterol at the interface between two aquaporin-0 (AQP0) tetramers. (**A**) Schematic figure illustrating how the insertion depth, *d*, and orientation angle, *θ*, of the cholesterol were measured. The cholesterol insertion depth was defined as the distance in *z* direction of the cholesterol oxygen atom (red stick representation) from the center of mass of the phosphorus atoms of the nearby sphingomyelin (SM) molecules in the extracellular leaflet (top green horizontal line). The cholesterol orientation was defined as the angle between the membrane normal (simulation box *z*-vector) and the vector along the rings of the cholesterol molecule (black dashed line). (**B–E**) Right panels: The three different systems that were simulated, namely (**B**) 'No AQP0', a pure lipid membrane without AQP0, (**C, D**) 'AQP0+EC lipids S2/1', a membrane with one AQP0 tetramer surrounded by the annular lipids seen in the AQP0$_{1SM:2Chol}$ structure, and (**E**) '2×AQP0', a membrane containing a pair of AQP0 tetramers together with the lipids in between them from a hybrid AQP0$_{2SM:1Chol}$ structure that replaces the two central SM molecules with the EC deep cholesterol molecules found in the AQP0$_{1SM:2Chol}$ structure. Left panels: Graphs showing normalized histograms for the insertion depth, *d*, and orientation angle, *θ*, for the monitored deep cholesterol in membranes with different SM:Chol ratios (see color code in panel **A**), except for the 2×AQP0 system, which was only simulated in a pure SM membrane. For the simulations with one AQP0 tetramer, insertion and orientation were computed separately for the deep cholesterol located at surface S2 (**C**) and S1 (**D**). The vertical line indicates the most probable cholesterol position in the 2×AQP0 system (**E**). See also *Figure 6—figure supplement 1*.

The online version of this article includes the following figure supplement(s) for figure 6:

**Figure supplement 1.** Time trace of the insertion depth of cholesterol.

## Deep-binding cholesterol occurs at the interface between two AQP0 tetramers

To assess the behavior of the deep cholesterol associated with AQP0 in terms of orientation and membrane-insertion depth (*Figure 6A*), we analyzed several different situations (*Figure 6C–E*). For this analysis, we initially positioned a cholesterol molecule in the middle of the bilayer according to the relative observed position of the deep cholesterol (Chol3) in the AQP0$_{1SM:2Chol}$ structure. For reference, we also simulated a cholesterol inserted at the equivalent position into a pure lipid membrane (*Figure 6B*). The presence of cholesterol in the membrane did not significantly alter the insertion depth of the monitored cholesterol. Within a few ns, cholesterol moved away from the deeply inserted position and equilibrated to a stable position with an insertion level of –0.73±0.31 nm (avg. ± s.d.) in a pure SM bilayer and –0.58±0.16 nm (avg. ± s.d.) and –0.60±0.16 nm in the 2:1 and 1:2 SM:Chol membranes, respectively (*Figure 6B* and *Figure 6—figure supplement 1*). In terms of orientation, an increasing cholesterol concentration in the membrane resulted in a decreasing tilt of the monitored cholesterol as well as in a narrower distribution (*Figure 6B*).

The behavior of the monitored deep cholesterol was very different when it was associated with AQP0. In this case, the protein becomes the main factor that defines its insertion depth and orientation (*Figure 6C and D*). The absence or presence of cholesterol in the membrane has no effect on the behavior of the AQP0-associated deep cholesterol: this sterol moved very quickly away from the initial position (*Figure 6—figure supplement 1*) and stabilized at around –0.6 nm which is a similar value to that seen in SM membranes without AQP0 but less than that of the deep cholesterol seen in the AQP0$_{1SM:2Chol}$ structure where it is close to –1.0 nm (*Figure 6C and D*). In terms of orientation, the AQP0-associated deep cholesterol sampled tilt angles up to ~70° with a distribution rather insensitive to the cholesterol concentration in the membrane (*Figure 6C and D*). Moreover, it appears that both surfaces of AQP0, S1 and S2, have a similar effect on the insertion depth and orientation of the associated deep cholesterol (compare *Figure 6C and D*).

The behavior of the deep cholesterol changed again when it was sandwiched in between two neighboring AQP0 tetramers. While the tilt distribution of the cholesterol is comparable to when it interacts only with one AQP0 tetramer, although it may be able to adopt slightly larger tilts, the insertion depth is distinctly different (*Figure 6E*). Here, the average insertion depth of –1.16±0.50 nm is now similar to the position of Chol3 in the AQP0$_{1SM:2Chol}$ structure of –1.0 nm.

Note that we observed a few spontaneous cholesterol flip-flop events, as previously observed in other microsecond-scale simulations (*Gu et al., 2019*; *Jo et al., 2010*; *Marino et al., 2016*), resulting in this particular case in the small peaks at ~–3.25 nm insertion and ~135° orientation in *Figure 6E* (see also the sudden drop in the insertion time trace in *Figure 6—figure supplement 1*).

In summary, our simulations show that the association of a cholesterol with an individual AQP0 tetramer defines the extent of its insertion depth and orientation, overriding any effect that the cholesterol concentration of the surrounding membrane may have. Furthermore, the association of a cholesterol with two adjacent AQP0 tetramers induces it to localize much deeper in the membrane than otherwise observed and supports the position of Chol3 seen in the AQP0$_{1SM:2Chol}$ structure.

## Sandwiched cholesterol increases the force needed to separate adjacent AQP0 tetramers

Considering the different behavior of cholesterol associated with one AQP0 tetramer *versus* being sandwiched in between two adjacent tetramers, we next asked whether the presence of cholesterol in between two AQP0 tetramers could have an effect on the stability of their association. We considered two lipid interfaces between the two neighboring tetramers: an 'SM interface' that consisted of only SM lipids at the positions seen in the AQP0$_{2SM:1Chol}$ structure, and a second identical one, except that two SM molecules were replaced by the deep-binding Chol3 molecules in the position observed in the AQP0$_{1SM:2Chol}$ structure (*Figure 5—figure supplement 1D*). Comparison of these two cases should allow us to assess the effect of the deep-binding Chol3 molecules on the mechanical stability of the associated AQP0 tetramers. During a cumulative time of 10 μs of equilibrium MD simulations, the two AQP0 tetramers overall behaved very similarly for the two interfaces and stayed close together (*Figure 7A*), exhibiting only minor variations (of less than 13.4°) in angular lateral, tilt, and torsional intra-tetramer motions (*Figure 7B*). To explore the effect of the deep cholesterols on the mechanical stability of the paired tetramers in response to a lateral mechanical perturbation, we applied an

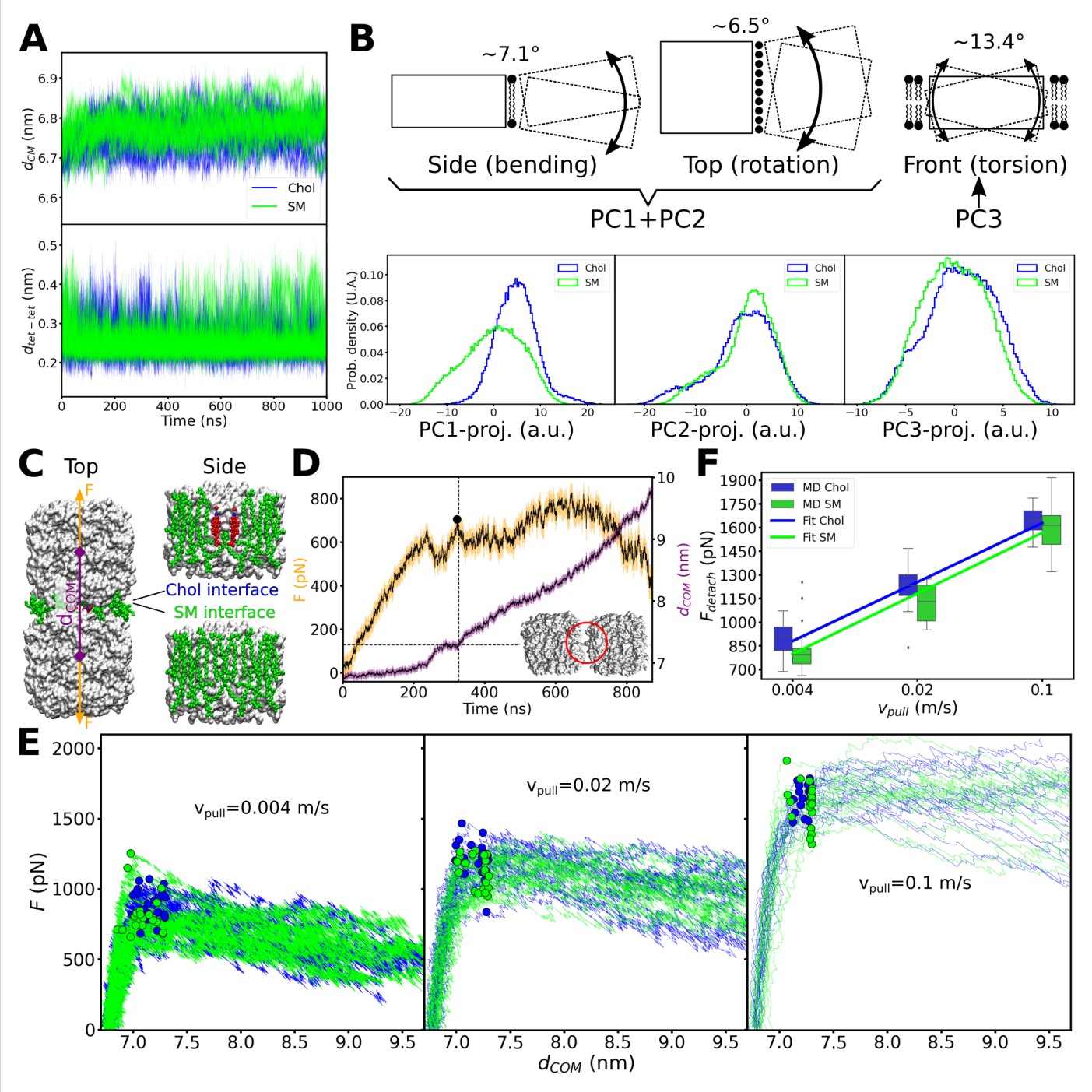

**Figure 7.** Equilibrium molecular dynamics (MD) simulations of pairs of associated aquaporin-0 (AQP0) tetramers and force-induced separation of two associated AQP0 tetramers. (**A**) Distance between the centers of mass $d_{CM}$ (top) and minimum distance $d_{tet\text{-}tet}$ (bottom) between the pair of AQP0 tetramers during the 2×AQP0 simulations in equilibrium for the interface containing only sphingomyelin (SM) molecules (SM, green) and the interface containing the deep cholesterol (Chol, blue) (n=10 simulations for each case). (**B**) Principal component (PC) analysis of the relative movements between the two tetramers. Here, the motion of one of the tetramers (dashed-line rectangle) relative to the other tetramer (solid-line rectangle) was monitored. The three main PC accounted for 67.9% of the total relative motion between tetramers (PC1: 34.3%, PC2: 24.7%, and PC3: 8.9%). The schematic drawings illustrate the three main modes of motion: bending (depicted in the drawing as viewed from the side of the membrane), lateral rotation (depicted in the drawing as viewed from the top of the membrane), and torsion (depicted in the drawing as viewed from the side of the membrane with one tetramer in front of the other). Lipids at the interface between the two tetramers (first two panels) and lipids surrounding the two tetramers (last panel) are shown. PC1 plus PC2 capture the bending and rotation while PC3 corresponds to torsion. The histograms in the bottom panels show the projections of the MD trajectories onto the three main PC vectors, for the interfaces containing only sphingomyelin (SM, green) or containing the

*Figure 7 continued on next page*

*Figure 7 continued*

deep cholesterols (Chol, blue). The approximate angular extent for each of the modes, attributed to these projections, is indicated (in degrees). The distributions with and without cholesterol are very similar, except for PC1. Nevertheless, PC1 relates to a small angular variation (~7.1° bending together with ~6.5° rotation). (**C**) Two AQP0 tetramers (white surface) arranged as in two-dimensional (2D) crystals and embedded in a pure SM membrane were pulled apart by exerting a harmonic force $F$ on them in the direction that connects their centers of mass ($d_{COM}$). Two different interfaces were studied: the 'SM interface' consisted solely of SM lipids (green spheres) as seen in the AQP0$_{2SM:1Chol}$ structure, whereas the 'Chol interface' contained the deep cholesterols seen in the AQP0$_{1SM:2Chol}$ structure. The reference position of the harmonic springs used to exert the force $F$ was moved at a constant velocity $V_{pull}$/2. (**D**) Force ($F$) and distance ($d_{COM}$) time traces are shown for one of the simulations using a $V_{pull}$ of 0.004 m/s. A Gaussian smoothing function (black continuous lines) was applied to the curves (yellow and purple). The detachment force (black circle) was computed as the highest recorded force when $d_{COM}$ started to increase and below a cut-off distance $d_{COM}$ of 7.3 nm (horizontal dashed line indicates the cut-off distance and the vertical dashed line indicated the time when this value was surpassed). Note that using different cut-off distances did not change the overall trend (see ***Figure 7—figure supplement 1***). The inset shows an example of the arrangement of the tetramers at the moment of detachment. The red circle indicates the last contact. (**E**) Force–distance profiles for the two different interfaces are presented for the three indicated pulling velocities ($n$=20 for each case). Dots indicate the point of detachment. (**F**) Detachment force is presented as a function of the pulling rate for the two interfaces (box plots, $n$=20). A fit of the form $F_{detach} = A + B*\log(V_{pull})$ is shown to guide the eye with lines ($F_{detach}$ = [2164+232*log($V_{pull}$)]) for the 'Chol' interface and $F_{detach}$ = [2116+239*log($V_{pull}$)]) for the 'SM' interface. p-Values comparing the two datasets, separately for each pulling velocity, are 0.022 ($V_{pull}$ = 0.004 m/s), 0.015 ($V_{pull}$ = 0.02 m/s), and 0.262 ($V_{pull}$ = 0.1 m/s) (Mann–Whitney U test). Furthermore, a two-way ANOVA test, considering the three pulling velocities at once, retrieved a p-value of 0.003 for the lipid interface change (i.e. Chol vs. SM). See also ***Figure 7—figure supplement 1***.

The online version of this article includes the following figure supplement(s) for figure 7:

**Figure supplement 1.** Detachment force versus pulling rate for different $d_{cut-off}$ criteria.

external harmonic force to the tetramers, attaching virtual springs to their centers of mass and moving these springs in opposite directions at constant velocity (***Figure 7C***). The force exerted on the two tetramers was ramped up until the two tetramers detached from each other and the force leveled off (***Figure 7D and E***). The resulting force–distance profiles revealed that the tetramer pairs withstood a higher force before detachment when cholesterol was present at the interface (***Figure 7E and F***). More quantitatively, at our fastest pulling rate of 0.1 m/s, at which the integrity of the membrane was still maintained, the detachment forces were not significantly different whether the interface contained cholesterol or not (***Figure 7F***). However, for the slower pulling rates of 0.02 m/s and 0.004 m/s, which are more relevant, because there should not be strong forces pulling on AQP0 tetramers in native membranes, the detachment force required to separate the tetramers was statistically significantly higher when their interface contained cholesterol (***Figure 7F***). Thus, because a higher force is required to separate neighboring AQP0 tetramers when cholesterol is present at their interface, deep cholesterols appear to mechanically stabilize AQP0 arrays.

## Protein–protein interactions affect the stability of associated AQP0 tetramers

We performed a further analysis in an effort to understand why the presence of cholesterol at the interface would increase the force needed to separate associated AQP0 tetramers. For this purpose, we calculated density maps representing what components the surface of an AQP0 tetramer interacts with over time. In the absence of cholesterol, the AQP0 surface is completely covered by sphingomyelin in the hydrophobic region of the membrane and by water outside this region (***Figure 8A***, left column). As noted before, there are essentially no direct protein–protein interactions between the adjacent tetramers. When cholesterol was present at the interface, it interacted with AQP0 at the center of the membrane and remained mostly in place (***Figure 8A***, right column). While the area below the cholesterols toward the cytoplasmic surface is occupied by sphingomyelin as in the interface lacking cholesterol, the area above the cholesterols toward the extracellular surface is now covered with water (***Figure 8A***, right column), which is in close proximity to the cholesterol oxygen group (***Figure 8—figure supplement 1***), and now also forms direct protein–protein interactions with the adjacent tetramer (***Figure 8A***, right column, and ***Figure 8—figure supplement 2***). We further analyzed these interactions, by computing the overall fraction of time during which residue–residue contacts formed (***Figure 8B and C***) and their average lifetimes (***Figure 8D***). The overall fraction is a thermodynamic quantity that relates to the probability of formation of these contacts, while the lifetime is related to the kinetics of this process. The probability of formation increased mainly for residue pairs above the two cholesterol molecules (***Figure 8B and C***). The lifetime also slightly increased for

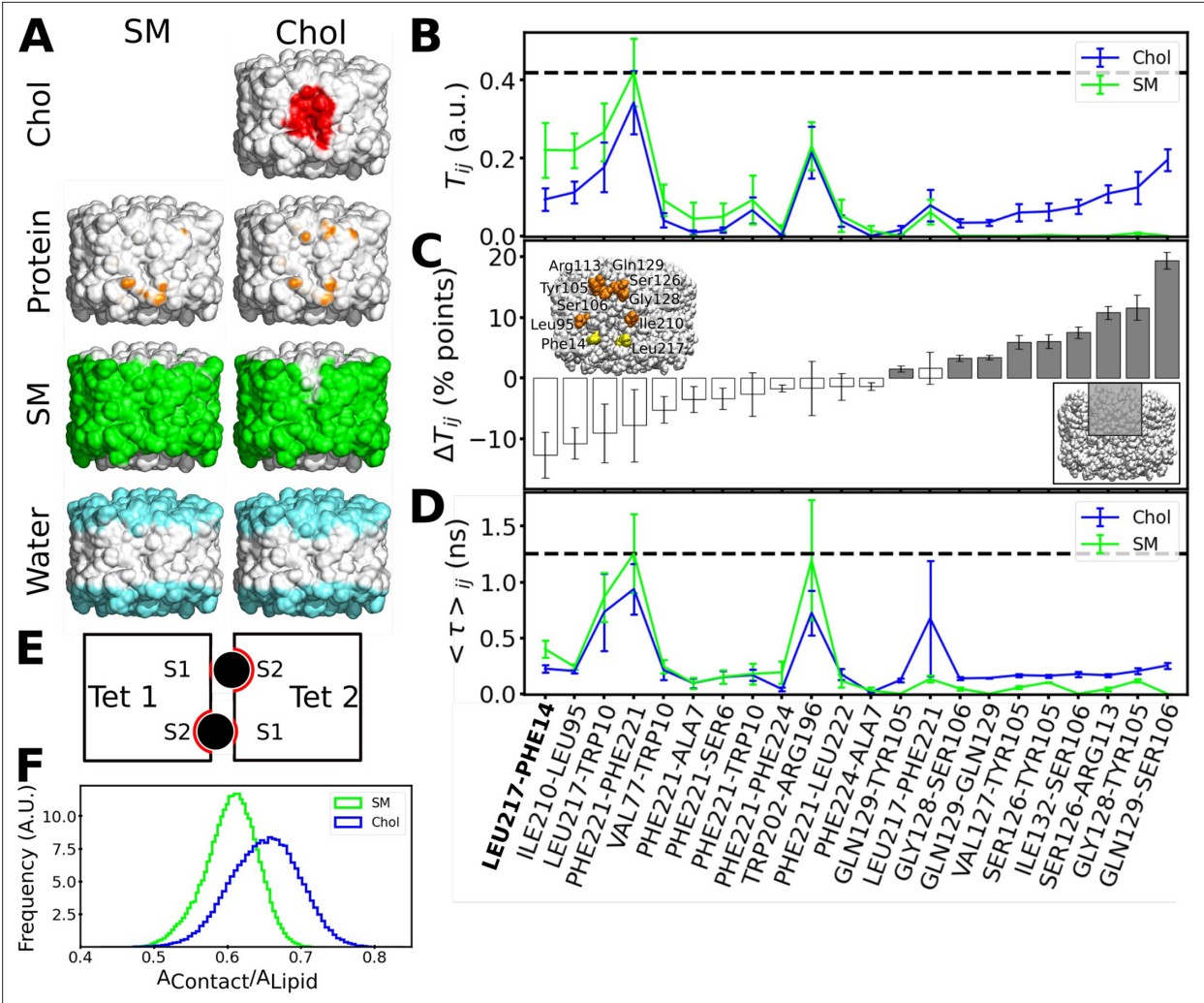

**Figure 8.** Interactions formed between adjacent aquaporin-0 (AQP0) tetramers and lipid-AQP0 surface complementarity. (**A**) Equilibrium molecular dynamics (MD) simulations were performed with pairs of AQP0 tetramers without cholesterol (left panels) or with cholesterol (right panels) at the interface. Density maps were calculated for different components next to the AQP0 surface over time and these maps were projected onto the surface of one of the AQP0 tetramers to represent their localization. For clarity, the second tetramer, which would be in front of the shown tetramer, is not shown. The density displayed at arbitrary density units is color-coded and shows the position of cholesterol (red), protein, i.e., the neighboring AQP0 tetramer (brown), sphingomyelin (green), and water (cyan). (**B**) $T_{ij}$ is defined as the fraction of time during which residues ($i$, $j$) from opposite tetramers are in contact. This quantity was extracted from the equilibrium simulations for the pure sphingomyelin interface $T_{ij}$(SM) and for the interface containing cholesterol $T_{ij}$(Chol). $T_{ij}$ = 0 means that $i$ and $j$ were never in contact and $T_{ij}$ = 1 means that they were always in contact. (**C**) The pairwise difference $\Delta T_{ij}$ = $T_{ij}$(Chol) - $T_{ij}$(SM) is shown, discarding insignificant changes ($\Delta T_{ij}$<1 percentage, %, points). Accordingly, a value of $\Delta T_{ij}$>0 ($\Delta T_{ij}$<0) corresponds to protein–protein contacts that were more often observed in the simulations with cholesterol (sphingomyelin). For instance, the residue pair Gln129–Ser106 was observed almost 20 percentage points more time in the simulations with cholesterol. The color of the bars indicates the location of the residues (gray: inner part of the extracellular leaflet, i.e., where deep cholesterol resides; white: the rest of the interfacial protein surface, see inset at lower right). Residues involved in a high $\Delta T_{ij}$>10 percentage points are highlighted in orange in the inset (yellow for the contact Leu217–Phe214 observed in the electron crystallographic structure). (**D**) The average duration for every established protein–protein contact, <$\tau$>$_{ij}$ is also displayed for the two different lipid interfaces. In (**B**) and (**D**), the horizontal dashed lines indicate the highest value observed for all possible residue pairs. Contacts observed in the electron crystallographic structures are highlighted in bold letters. In B–D, the avg ± s.e.m. is presented ($n$=20, i.e. 10 independent simulation times with two symmetric monomeric interfaces). (**E**) The schematic drawing depicts a top view of the two associated AQP0 tetramers (squares, 'Tet 1' and 'Tet 2') with the two central lipids sandwiched between them (black circles). The other lipids at the interface are not shown for clarity. The respective monomer surfaces S1 and S2 are indicated. The region of the surface of the two tetramers that is in total covered by a lipid, $A_{Contact}$, (red line) was normalized by the surface area of the lipid, $A_{Lipid}$ (here corresponding to the perimeter of the circles). This ratio gives a measure of the surface complementarity between the tetramers and the sandwiched lipids, i.e., the higher the value of $A_{Contact}$/$A_{Lipid}$ the more the two surfaces complement each other. (**F**) Normalized histograms of $A_{Contact}$/$A_{Lipid}$ obtained from the equilibrium MD simulations are shown for the central sphingomyelin (SM) and cholesterol (Chol). See also *Figure 8—figure supplements 1–3*.

*Figure 8 continued on next page*

*Figure 8 continued*

The online version of this article includes the following figure supplement(s) for figure 8:

**Figure supplement 1.** Distance from each deep cholesterol OH group to the nearest water molecule.

**Figure supplement 2.** Contacts between tetramers in 2×AQP0 simulations.

**Figure supplement 3.** Surfaces of aquaporin-0 (AQP0) and AQP1.

the residue pairs above the cholesterol molecules, but these contacts occurred mainly among the highly flexible amino acid side chains and were short-lived, i.e., smaller than 1 ns (*Figure 8D*).

Only one inter-tetramer contact was observed in the EC structure of AQP0 in a high cholesterol membrane, AQP0$_{1SM:2Chol}$, namely Leu217$_{Tet1}$–Phe14$_{Tet12}$. This contact was also observed in our simulations (*Figure 8B–D*). However, the AQP0$_{1SM:2Chol}$ structure did not reveal protein–protein contacts above the cholesterol molecules (*Figure 8B–D*). A possible explanation for this discrepancy is that although such contacts could be observed in our simulations for the cholesterol-containing interface, they had a rather low probability to occur (only up to 20% of the simulated time, *Figure 8C*). Thus, the EC structure would visualize other contacts with a higher probability, in particular protein–water interactions (*Figure 8A*, right column).

## Lipid–protein surface complementarity may also enhance mechanical resistance of associated AQP0 tetramers

We found that the presence of cholesterol at the interface results in new protein–protein interactions between adjacent AQP0 tetramers that may increase the stability of their association, but we wondered whether the cholesterol itself could contribute to the increased force needed to separate associated tetramers. Surface complementarity has previously been suggested to play a key role in modulating lipid–protein interactions (*Aponte-Santamaría et al., 2012*; *Niemelä et al., 2010*) with particular relevance for cholesterol (*Kurth et al., 2020*), prompting us to analyze the surface complementarity between AQP0 and cholesterol. We quantified surface complementarity as the contact area between the sandwiched lipids (either cholesterol or sphingomyelin) and the two tetramers, $A_{Contact}$, and normalized it by the surface area of the lipid in question, $A_{Lipid}$, i.e., a larger $A_{Contact}/A_{Lipid}$ ratio would indicate a higher surface complementarity between the lipid and the proteins (see Materials and methods for details of this calculation and *Figure 8E* for a schematic diagram). We calculated the $A_{Contact}/A_{Lipid}$ ratio from the equilibrium simulations. Remarkably, the sandwiched cholesterol displayed a higher surface complementarity than the sandwiched sphingomyelin (*Figure 8F*). This analysis thus suggests that cholesterol better accommodates to the roughness of the two AQP0 tetramer surfaces than sphingomyelin can, a feature that may contribute to the enhanced resistance of associated tetramers to detach when cholesterol is present at their interface.

## Discussion

In this study, we determined two EC structures of AQP0 in membranes formed by sphingomyelin and cholesterol. As these so-called raft lipids are the main constituents of lens membranes, they represent the natural environment of AQP0. The AQP0 crystals formed in sphingomyelin/cholesterol membranes have the same lattice constants as previous 2D crystals obtained with other lipids (*Gonen et al., 2005*; *Hite et al., 2010*) as well as the 2D arrays found in lens membranes (*Buzhynskyy et al., 2007*). The 2D crystals thus reflect the arrangement of AQP0 tetramers in native 2D arrays.

Our structure of AQP0$_{2SM:1Chol}$ reveals seven sphingomyelins (*Figure 3A*), the same number of lipids also seen in structures of AQP0 in bilayers formed by DMPC (*Gonen et al., 2005*) and EPL (*Hite et al., 2010*), and these are also located in almost identical positions (*Figure 3B and C*). The interaction of sphingomyelins with AQP0 thus seems to be governed by the same principles that were found for phosphoglycerolipids, namely that the acyl chains locate to grooves in the protein surface and that the lipid head groups make few if any interactions with the protein (*Hite et al., 2010*). However, compared to the acyl chains in the EPL bilayer, those of the sphingomyelins appear to be straighter, which likely reflects the saturated nature of the sphingomyelin acyl chains. Despite this difference, the structure of the sphingomyelin bilayer is very similar to those of bilayers formed by

phosphoglycerolipids, indicating that sphingomyelin by itself is unlikely the reason for AQP0 to form 2D arrays in lens membranes.

Cholesterol has a higher affinity for sphingomyelin than for phosphoglycerolipids (*Niu and Litman, 2002*), because the mostly saturated acyl chains of sphingomyelin can better accommodate the planar ring system of cholesterol (*Epand and Epand, 2004*; *Zheng et al., 2007*) and because the sphingo-myelin backbone has two hydrogen-bond donor groups, an amide and a hydroxyl group, that make hydrogen bonding to cholesterol more effective (*Róg and Pasenkiewicz-Gierula, 2006*). Furthermore, the interaction of the sphingomyelin amide group with the hydroxyl group of cholesterol orients the smooth (α) face of the cholesterol toward sphingomyelin, leaving only the rough (β) face to interact with membrane proteins (*Fantini and Barrantes, 2013*). In the context of AQP0 2D arrays, however, these interactions do not appear to occur. While Chol1 does interact with sphingomyelin acyl chains in both structures, the additional cholesterols in the AQP0$_{1SM:2Chol}$ structure seem to have little inter-actions with sphingomyelin acyl chains and instead seem to substitute for some of the sphingomyelin acyl chains seen in the AQP0$_{2SM:1Chol}$ structure (*Figure 4A and C*). We also do not observe interactions between the hydroxyl head group of any cholesterols with an amide group of the sphingomyelins. While we cannot rule out that specific sphingomyelin–cholesterol interactions occur in the membrane area enclosed by four AQP0 tetramers (where we were unable to build models for the lipids due to the fourfold symmetry axis), our structures suggest that specific sphingomyelin–cholesterol interactions do not play a critical role in AQP0 array formation in the lens membrane.

Cholesterol increases the order of lipid acyl chains (*de Meyer and Smit, 2009*; *Lafleur et al., 1990*), and the resulting phase transition is thought to contribute to the segregation of cholesterol-enriched membrane areas (*Pandit et al., 2007*). A potential mechanism by which cholesterol could induce AQP0 array formation could thus be that cholesterol bound to AQP0 would order the surrounding lipids, leading to a phase transition that leads to an initial segregation and crystallization of the AQP0/cholesterol/sphingomyelin units in the membrane.

In an attempt to assess this possibility, we looked at the *B*-factors of the acyl chains, which are affected, among other factors, by the mobility of the atoms. Comparison of corresponding acyl chains showed that the *B*-factors of the acyl chains in the AQP0$_{1SM:2Chol}$ structure tend to be slightly lower than those of the acyl chains in the AQP0$_{2SM:1Chol}$ structure (*Figure 4—figure supplement 2*), suggesting that the cholesterols could have an ordering effect on the sphingomyelin acyl chains. However, the *B*-factors of the acyl chains in the AQP0$_{EPL}$ structure, which contain double bonds and are not constrained by cholesterol and should thus be much less ordered, are similar to those of the acyl chains in the AQP0$_{2SM:1Chol}$ structure. A likely explanation is that acyl chains, including unsaturated ones, preferentially fill in grooves in the protein surface (*Hite et al., 2010*), which should already substantially constrain their mobility. Therefore, cholesterol bound to the protein surface may not add much additional constraints to their already restricted mobility. Cholesterol-induced phase separation thus does not seem to be a likely cause for AQP0 array formation.

The hydrophobic thickness is an important characteristic of a lipid bilayer, which depends on the length and saturation of the acyl chains of the lipids forming the bilayer. Mismatch of the hydrophobic thickness between a membrane protein and the lipid bilayer is thought to be one mechanism by which membrane proteins can cluster in a membrane (*Hanulová and Weiss, 2012*; *Schmidt et al., 2008*). One factor that affects the hydrophobic thickness of a membrane is its cholesterol content, which is related to the ordering effect of cholesterol on the lipid acyl chains (*de Meyer and Smit, 2009*; *Lafleur et al., 1990*). Change in cholesterol-induced hydrophobic thickness may thus be another potential force that drives the formation of AQP0 arrays in the lens membrane. Our measurements of the average distance between the phosphorus atoms of the phosphodiester groups and the nitrogen atoms of the amide groups of the sphingomyelins in the two leaflets (*Figure 4A and B*) show that both distances are larger for the bilayer with the higher cholesterol content, by ~3 Å for the phosphodiester groups and ~4 Å for the amide groups, suggesting that a higher cholesterol content indeed increases the hydrophobic thickness of the bilayer formed by the annular sphingomyelins.

In cells, sphingomyelin is predominantly found in the extracellular leaflet of the plasma membrane (*Devaux, 1991*), so that sphingomyelins seen in the extracellular leaflet in our two structures likely represent true positions of sphingomyelin in native AQP0 2D arrays. The two sphingomyelins in the extracellular leaflet that are seen in both sphingomyelin/cholesterol membranes are in exactly the same position (*Figure 4C*), establishing that the extracellular leaflet does not change its position

relative to AQP0 upon thickening of the membrane with increasing cholesterol content. Sphingo-myelin SM3 may be responsible for the fixed position of the extracellular leaflet. In all structures of AQP0 in different lipid bilayers, the lipid at the position equivalent to that of SM3 always has the best-defined density and the lowest $B$-factors (including for the acyl chains; *Figure 4—figure supplement 2*). The lipid at this position is always the only one whose head group makes interactions with AQP0. It is thus possible that the lipid at this position defines and locks in the position of the extracellular leaflet with respect to AQP0. As a result, the extracellular leaflet does not play a role in defining the thickness of the membrane.

The increase in hydrophobic thickness with higher cholesterol content is the result of the three sphingomyelins in the cytoplasmic leaflet moving further out from the bilayer center, which they do even though this leaflet contains only a single cholesterol (*Figure 4C*). The notion that an increase in hydrophobic thickness is predominantly due to movements of lipids in the cytoplasmic leaflet is consistent with the observation that these lipids are usually more mobile, i.e., have higher $B$-factors in crystal structure than lipids in the extracellular leaflet (*Belrhali et al., 1999*). Comparison of the lipid arrangement in the cytoplasmic leaflet shows that Chol4 in the AQP0$_{1SM:2Chol}$ structure assumes the position of SM6 in the AQP0$_{2SM:1Chol}$ structure (*Figure 4A–C*). The new position of the displaced SM6 in the AQP0$_{1SM:2Chol}$ structure is further out from the bilayer center and potentially is the cause for similar outward movements of lipids SM5 and SM7, thus defining the new position of the cytoplasmic leaflet and increasing the hydrophobic thickness of the annular lipid bilayer. Hence, a higher cholesterol concentration of the membrane does increase the hydrophobic thickness of the lipid bilayer formed by the annular lipids surrounding AQP0 tetramers, thus creating a hydrophobic mismatch with the remaining lipid bilayer, which, in turn, would provide a driving force for AQP0 tetramers to cluster in the native lens membrane.

The cholesterols in our structures do not interact with AQP0 through consensus cholesterol-binding sites known as the Cholesterol Recognition/interaction Amino acid Consensus sequence (CRAC domain; -L/V-(X)$_{1-5}$-Y-(X)$_{1-5}$-R/K-) (*Li and Papadopoulos, 1998*) or the inverted CRAC domain, CARC (*Baier et al., 2011*). Cholesterol can also bind to membrane proteins in a fashion that does not involve a CRAC or CARC domain, as seen for example for the influenza M2 protein (*Elkins et al., 2018*; *Elkins et al., 2017*) and α-synuclein (*Fantini et al., 2011*). Emerging cryo-EM structures of G-protein-coupled receptors (GPCRs) also do not show a consensus motif for their interaction with cholesterol (*Sarkar and Chattopadhyay, 2020*; *Taghon et al., 2021*). Furthermore, the cholesterols interact with AQP0 through their smooth α face (*Figure 4—figure supplement 1*), which is not engaged in interactions with sphingomyelin as previously thought (*Fantini and Barrantes, 2013*), thus differing from the predominant interaction of cholesterols with GPCRs, the G-protein-gated inwardly rectifying potassium-2 (GIRK2) channel, and the Na$^+$, K$^+$-ATPase, which occur predominantly through the rough β face of cholesterol (*Gimpl, 2016*; *Mathiharan et al., 2021*; *Shinoda et al., 2009*). However, there are other cases in which cholesterol interacts with membrane proteins through its α face, such as seen for the Niemann–Pick C1-like 1 protein (*Huang et al., 2020*). Likely the residues on the protein surface determine whether a protein interacts with cholesterol through its α or β face. The smooth α face interacts with aromatic side chains through π-stacking (*Nishio et al., 1995*), whereas the β face appears to interact with hydrophobic side chains, such as leucine, valine, or isoleucine (*Fantini et al., 2011*).

Chol3 is the most unusual lipid in our structures, because it is located in the middle of the lipid bilayer and because it is sandwiched between two neighboring AQP tetramers. Binding sites for cholesterol in the middle of the membrane have also been described for GPCRs, and these sites have been named 'deep binding sites' (*Genheden et al., 2017*). A docking investigation with known membrane protein structures for cholesterol-binding sites revealed deep binding sites not only in GPCRs but also in ion channels and transporters (*Lee, 2018*). Unlike any other cholesterol in our structures, Chol3 interacts directly with two AQP0 subunits that are part of different tetramers. This interaction is different from the cholesterol-induced dimerization of GPCRs as these are mediated by interactions between two or more cholesterol molecules rather than one cholesterol directly interacting with two GPCRs (*Gimpl, 2016*; *Hanson et al., 2008*). The two adjacent AQP0 tetramers form a pocket that is ~8.5 Å wide at the position of the ring system and ~2.5 Å at the position of the acyl chain (*Figure 4E*). The interaction surface on the AQP0 subunit interacting with the smooth face of Chol3 encompasses 373 Å$^2$ and is formed by residues Ser106, Val103, Ala102, Ala99, and Leu21, which line the ring system, and three phenylalanine residues, Phe17, Phe18, and Phe14, which surround the

acyl chain. The interaction surface on the AQP0 subunit interacting with the rough face of Chol3 is 344 Å$^2$ and is predominantly formed by residues Ile132, Ile135, Phe136, Ile210, Leu214, and Leu217 (*Figure 4E*). Leucine and isoleucine residues have also been found to mediate the interaction of many GPCRs with the rough β face of cholesterol (*Gimpl, 2016*). The hydroxyl head group of Chol3 interacts with a water molecule that may be stabilized through an interaction with Ser126 (*Figure 4E*). Because of its intricate interactions with two AQP0 subunits, Chol3 can likely function as a glue that can keep two tetramers together. This structural characteristic may be the reason why AQP0 can form 2D arrays in pure cholesterol membranes (*Figure 1E and F*).

To gain further insights into the role cholesterol plays in the stability of AQP0 arrays, we used MD simulations to investigate the interplay between cholesterol and AQP0. The first set of simulations investigated the localization of cholesterol around a single AQP0 tetramer. Cholesterol positioned at well-defined positions, i.e., hotspots, that were in notable agreement with both the predicted positions from a previous computational study (*O'Connor and Klauda, 2011*) and the positions in the EC structures (*Figure 5*). Therefore, as it has been previously demonstrated to be the case for phospholipids (*Aponte-Santamaría et al., 2012*), the crystal packing in the crystallographic structures does not greatly affect the positioning of cholesterol around AQP0. In consequence, crystallographic cholesterol positions recapitulate the behavior of unconstrained cholesterol molecules around a single AQP0 tetramer. This is the case for all cholesterols except for one, the deep cholesterol, which will be discussed in detail below. Cholesterol localization around membrane proteins has been extensively studied, especially in the context of the regulation and activity of GPCRs, exchangers, and ion channels, but also aquaporins (for a comprehensive review, see *Corradi et al., 2019*). Interestingly, in coarse-grained MD simulations, when AQP1 was embedded in a lipid bilayer resembling the plasma membrane, cholesterol was enriched at two positions that qualitatively correspond to the lateral projections of Chol3 and Chol4 positions observed here for AQP0 (compare *Figure 5* with Figure 2 of *Corradi et al., 2018*). This suggests that individual AQP0 and AQP1 tetramers share similarities in their cholesterol-binding fingerprints, as was already shown for phospholipids in this protein family (*Stansfeld et al., 2013*). Nevertheless, cholesterol was rather depleted from the Chol1 position in AQP1. The shape of the protein surface has been shown to critically define the positions where lipids non-specifically associate with AQP0 (*Aponte-Santamaría et al., 2012*; *Briones et al., 2017*) and also other proteins (e.g. ion channels [*Niemelä et al., 2010*] and GPCRs [*Kurth et al., 2020*], among many others reviewed in *Corradi et al., 2019*). This principle presumably applies here, too, i.e., the different surface of AQP0 at the Chol1 position, compared to that of AQP1, may result in a greater association of cholesterol over sphingolipids in this position (*Figure 8—figure supplements 1 and 3*).

In a second set of simulations, we attempted to understand the molecular reasons for the unusual position of deep-binding Chol3 (*Figure 6*). We demonstrate that neither a single AQP0 tetramer (*Figure 6C and D*) nor a specific bulk cholesterol concentration (*Figure 6B*) is sufficient to maintain annular cholesterol molecules at such a deeply inserted position. Instead, the constraint imposed by two adjacent AQP0 tetramers and the coordination of two water molecules were required (*Figures 6E and 8A* and *Figure 8—figure supplement 1*). Our data are thus consistent with the existence of deep cholesterol positions but only in a highly constrained setup that involves two AQP0 tetramers. The immediate questions raised by the existence of deep cholesterols are whether these affect the formation and/or stability of 2D AQP0 arrays.

Our simulations provide strong evidence for a direct influence of deep cholesterol molecules on the mechanical stability of the association between two adjacent AQP0 tetramers. We demonstrate that two adjacent tetramers can withstand larger lateral detachment forces when their interface contains deep cholesterol molecules (*Figure 7C–F*). Crystal structures of several membrane proteins already revealed that cholesterol can mediate crystal packing interactions (*Crnjar and Molteni, 2021*; *Khunweeraphong et al., 2020*; *Wu et al., 2014*). Moreover, the presence of cholesterol has been reported to influence membrane–protein oligomerization, a feature mainly explored for GPCRs (*Corradi et al., 2019*; *Gahbauer and Böckmann, 2016*). We expand on these data, by providing evidence of cholesterol—directly—engaging in the stronger attachment of two membrane proteins. We attribute this to cholesterol facilitating the formation of transient protein–protein contacts (*Figure 8A–D*) together with providing higher surface complementarity when sandwiched between two AQP0 tetramers (*Figure 8E and F*). Cardiolipin is a phospholipid that has been implicated in stabilizing transient α-helical membrane protein oligomers (*Gupta et al., 2017*). More specifically, cardiolipin has been

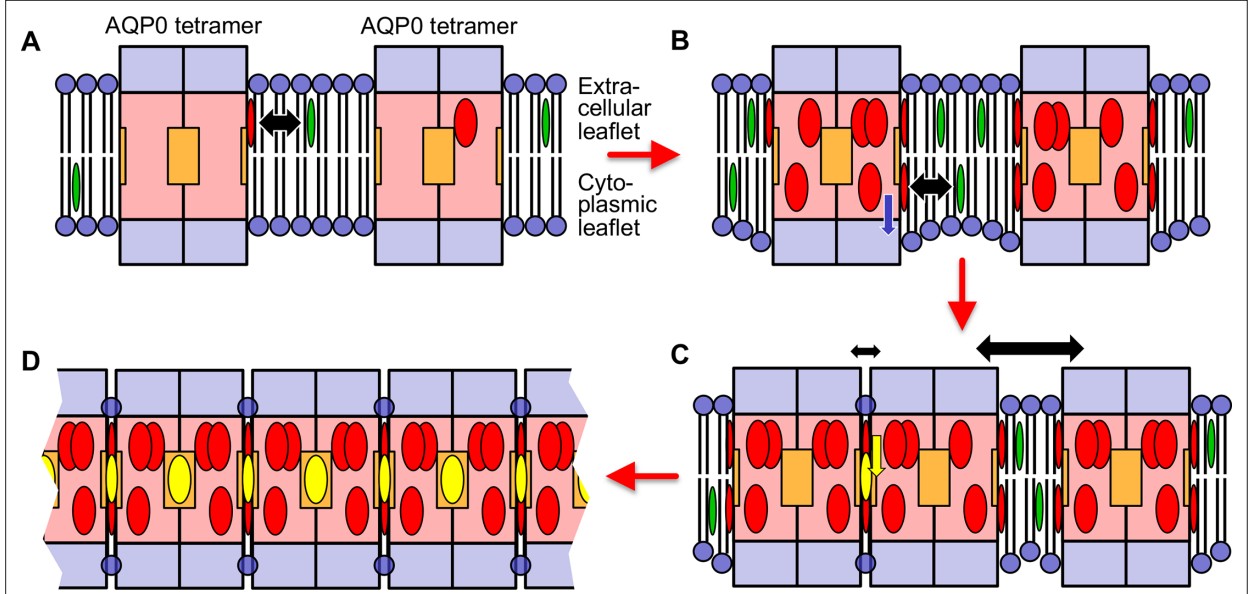

**Figure 9.** Proposed model for how an increasing cholesterol concentration drives aquaporin-0 (AQP0) two-dimensional (2D) array formation in the native lens membrane. (**A**) At a low cholesterol concentration, AQP0 tetramers are mostly surrounded by phospholipids and sphingomyelin. Free cholesterol in the membrane (green ovals) only associates with the highest affinity cholesterol-binding sites. Cholesterols occupying these peripheral binding sites are shown as red ovals and the black double-headed arrow indicates the transient nature of this interaction. The deep cholesterol-binding sites (orange squares) are not occupied. (**B**) With increasing cholesterol concentration, more cholesterols associate with the AQP0 surface. These cholesterols cause the interacting lipids in the cytoplasmic leaflet to move out from the bilayer center (blue arrow), resulting in the annular lipid shell that has a bigger hydrophobic thickness than the surrounding membrane, creating a hydrophobic mismatch that results in membrane deformation. (**C**) To minimize hydrophobic mismatch, AQP0 tetramers cluster. Cholesterol in between adjacent tetramers can move into the deep binding sites (yellow arrow) and cholesterol occupying deep binding sites (yellow ovals) act as glue that increases that association of the adjacent tetramers (indicated by the small double-headed black arrow) as compared to adjacent tetramers that do not sandwich a deep-binding cholesterol (indicated by the large double-headed black arrow). Clustering of proteins to minimize hydrophobic mismatch and stabilization by deep cholesterol-mediated protein–protein interactions may be the basis for the formation of transient lipid rafts. (**D**) Each AQP0 tetramer has four deep cholesterol-binding sites. As a result of the avidity effect, AQP0 can form large and stable 2D arrays.

described to serve as a 'glue' for the cytochrome components of the respiratory chain in mitochondrial membranes (*Arnarez et al., 2013*; *Zhang et al., 2002*). Interestingly, cardiolipin also impacts the resistance of AQPZ, the bacterial homolog of AQP0, to unfolding (*Laganowsky et al., 2014*). In addition, lipids have been recently shown to mediate protein assembly in the outer membrane of bacteria (*Webby et al., 2022*). Analogous to these systems and based on our EC structures and MD simulations, we propose that deep cholesterol increases the mechanical stability of associated AQP0 tetramers. Although we provide solid evidence here that deep cholesterol imparts mechanical stabilization, free energy calculations would be required to obtain the full picture of thermodynamic stabilization. However, such free energy calculations are challenging for lipids, due to the chemical complexity and poor convergence involved (*Wang et al., 2021*), and are thus beyond the scope of the current work.

Our structures of AQP0 arrays in sphingomyelin bilayers with low and high cholesterol content and the results from the MD simulations allow us to propose a model for cholesterol-induced array formation (*Figure 9*). In a lipid bilayer with a low cholesterol concentration, cholesterol may interact with AQP0 predominantly through the most specific cholesterol-binding site, the one occupied by Chol1, but this interaction would be transient and would not have a meaningful effect on the organization of the surrounding lipid molecules (sphingomyelin in our AQP0$_{2SM:1Chol}$ structure) (*Figure 9A*). An increase in cholesterol concentration will cause more cholesterols in the extracellular leaflet to bind to AQP0. In addition, cholesterol will also enrich in the cytoplasmic leaflet and associate with AQP0. This interaction will shift the annular lipids in this leaflet further away from the membrane center and create a hydrophobic mismatch between the shell of annular lipids and the surrounding lipid bilayer (*Figure 9B*). Driven by the force to minimize membrane tension induced by hydrophobic mismatch,

AQP0 tetramers will cluster. Cholesterols in between the tetramers can diffuse deeper into the membrane and associate with deep binding sites on AQP0, thus stabilizing the interaction between adjacent tetramers (*Figure 9C*). Alternatively, it may also be possible that cholesterol first interacts with a deep binding site on one AQP0 but is only stabilized as the second AQP0 traps it in position. As each AQP0 tetramer has four deep cholesterol-binding sites, the avidity effect would result in the formation and stabilization of large 2D arrays as those seen in native lens membranes (*Figure 9D*).

This model is specific for the formation of AQP0 arrays in lens membranes, but we speculate that similar principles may underlie the organization of lipid rafts. AQP0 may be special in that it forms tetramers and thus has four deep cholesterol-binding sites, so that the avidity effect allows it to form much larger domains than seen for typical lipid rafts (*Simons and Ikonen, 1997*; *Zelenka, 1984*). In addition, it is presumably not common that the same protein has surfaces that can interact with both the smooth α and the rough β face of cholesterol. However, cholesterol would have the potential to mediate the association of any protein with an α face-interacting surface with any other protein that features a β face-interacting surface. Thus, while hydrophobic mismatch and phase separation may be driving forces that bring proteins into close proximity, cholesterol may be the actual glue that increases the time they remain associated.

# Materials and methods

## Purification of AQP0

AQP0 was purified as described in *Gonen et al., 2004*. Briefly, dissected cores of sheep lenses (Wolverine Packing Company, Detroit, MI, USA) were homogenized, and isolated membranes were sequentially washed with 10 mM Tris-HCl, pH 8.0, 4 M urea, and 20 mM NaOH, and then solubilized with 4% *n*-octyl-β-D-glucopyranoside (OG; Anatrace). Solubilized AQP0 was purified using anion-exchange (MonoQ; GE Healthcare) and size-exclusion (Superose-6; GE Healthcare) chromatography.

## 2D crystallization of AQP0

Purified AQP0 in 1.2% OG, 10 mM Tris-HCl (pH 8.0), and 100 mM NaCl was mixed at an LPR (wt/wt) of 0.2 with different mixtures of OG-solubilized sphingomyelin (*N*-palmitoyl-D-erythro-sphingosylphosphorylcholine) (Avanti) and cholesterol (Avanti). The mixtures were placed into dialysis buttons, and the detergent was removed by dialysis at 37°C against 10 mM MES (pH 6.0), 300 mM NaCl, 30 mM $MgCl_2$, and 0.05% $NaN_3$ for 1 week with daily buffer exchanges. The 2D crystal samples were prepared by negative staining with uranyl formate and assessed on a Philips CM10 electron microscope.

## Imaging and image processing of AQP0 2D crystals grown with pure cholesterol

$AQP0_{Chol}$ 2D crystals were prepared on molybdenum grids using the carbon sandwich method (*Gyobu et al., 2004*) and a trehalose concentration ranging from 3% to 5% (wt/vol). After blotting away excess trehalose solution with a filter paper, grids were quick-frozen in liquid nitrogen and transferred onto a cryo-specimen stage for EM data collection.

Data of untilted 2D crystals were collected with a Polara electron microscope (FEI Company, Hillsboro, OR, USA) operated at an acceleration voltage of 300 kV and equipped with a Gatan K2 Summit direct electron detector camera (Gatan, Pleasanton, CA, USA), using low-dose procedures, a calibrated magnification of ×50,926, and a defocus ranging from −500 nm to −1500 nm. Dose-fractionated images were recorded in super-resolution mode at a counting rate of 8 counts/pixel/s (8.33 counts/$Å^2$/s). Frames were read out every 150 ms and 16 frames were collected, resulting in an exposure time of 2.4 s and a total dose of 20 e⁻/$Å^2$. Motion-corrected sum images were generated using the program MotionCorr (*Li et al., 2013*).

Images of $AQP0_{Chol}$ were computationally unbent and corrected for the effects of the contrast transfer function using the 2dx software (*Gipson et al., 2007b*). The plane group symmetry of the projection map was analyzed with ALLSPACE (*Table 1*; *Valpuesta et al., 1994*). Fifteen images were merged using 2dx_merge (*Gipson et al., 2007a*), resulting in a projection map at 3.2 Å resolution. The phase residuals are listed in *Table 2*.

### Collection of electron diffraction data

2D crystals grown at molar sphingomyelin:cholesterol ratios of 1:2 and 2:1 were prepared by trehalose embedding as described above.

Electron diffraction patterns of AQP0 2D crystals were recorded with a Polara electron microscope (FEI Company, Hillsboro, OR, USA) operated at an acceleration voltage of 300 kV and equipped with a 4k × 4k CCD camera (Gatan, Pleasanton, CA, USA). The camera length was set to 1.9 m and a C2 aperture with a diameter of 30 μm was selected. The selected target areas on the grids were exposed for 30 s, corresponding to a total electron dose of approximately 10 electrons/Å$^2$. Diffraction patterns were collected at tilt angles of 0°, 20°, 45°, 60°, 65°, and 70°, and details are provided in *Table 3*.

### Diffraction data processing and model building

The graphical user interface of the IPLT diffraction processing software was used to index electron diffraction patterns (*Schenk et al., 2013*). Diffraction patterns that showed multiple 2D lattices were discarded. After subtracting the background resulting from inelastic scattering, the intensities of the reflections were extracted, integrated according to their 2D Miller indices, and then merged into reciprocal lattice lines as described before (*Gonen et al., 2004*). The reconstructed 3D lattice lines were then iteratively refined against the experimental data enforcing a *p*422 plane symmetry. The refined lattice lines were sampled along the *z** direction using the 'truncate' program in the CCP4 software package (*Winn et al., 2011*), assuming a crystal thickness of 200 Å. The dataset was phased by molecular replacement in PHASER (version 2.1) (*McCoy et al., 2007*), using as search template the AQP0$_{DMPC}$ structure (PDB code: 2B6O) but without the loops and C-terminal domain of AQP0 and without the DMPC lipids. The density map obtained with data recorded from AQP0 2D crystals formed with a molar sphingomyelin:cholesterol ratio of 2:1 allowed building of AQP0 residues Ser6 to Pro225 as well as seven sphingomyelins and one cholesterol. For the density map obtained with AQP0 crystals formed with a sphingomyelin:cholesterol ratio of 1:2, AQP0 could also be built from Ser6 to Pro225, and five sphingomyelin and four cholesterol molecules could be built. Models were built in Coot (version 0.8.2) (*Emsley and Cowtan, 2004*), and the topologies and geometry restraint parameters for cholesterol and sphingomyelin were generated using the eLBOW program (*Moriarty et al.,*

**Table 4.** Summary of the simulations.

| Scheme | Protein system | Membrane system | Pulling rate (m/s) | Number of replicas | Length per replica (ns) |
|---|---|---|---|---|---|
| | | Pure SM | | | |
| | | 2:1 SM:Chol | | | |
| | No AQP0 | 1:2 SM:Chol | | 5 | 500 |
| | | Pure SM | | | |
| | | 2:1 SM:Chol | | | |
| | One AQP0 tetramer without EC lipids | 1:2 SM:Chol | | 5 | 500 |
| | | 2:1 SM:Chol | | | |
| | One AQP0 tetramer with EC lipids | 1:2 SM:Chol | | 5 | 500 |
| | Two AQP0 tetramers with deep cholesterol | Pure SM | | | |
| Equilibrium simulations | Two AQP0 tetramer without deep cholesterol | Pure SM | N/A | 10 | 1000 |
| | | | 0.1 | | ~55 |
| | | | 0.02 | | ~230 |
| | Two AQP0 tetramers with deep cholesterol | Pure SM | 0.004 | 20 replicas for each velocity | ~900 |
| | | | 0.1 | | ~55 |
| | | | 0.02 | | ~230 |
| Force probe | Two AQP0 tetramers without deep cholesterol | Pure SM | 0.004 | 20 replicas for each velocity | ~900 |

*2009*). The model was refined in CNS (version 1.3) (*Brünger et al., 1998*) and Phenix (version 1.20.1) (*Adams et al., 2010*). Refinement statistics are summarized in *Table 3*. Figures were generated with PyMOL (version 1.8) (The PyMOL Molecular Graphics System, Schrödinger, LLC) and UCSF Chimera (version 1.10) (*Pettersen et al., 2004*).

## MD simulations under equilibrium conditions

The simulations performed under equilibrium conditions are summarized in *Table 4*. For simulations of pure lipid bilayers, a single cholesterol molecule was inserted in a membrane of 128 lipids at the same height, relative to the neighboring lipids, as the deep cholesterol seen in the AQP01SM:2Chol structure (*Figure 5—figure supplement 1A*). For simulations using a single AQP0 tetramer, the tetramer was inserted into a membrane containing ~488 lipids, either by itself (*Figure 5—figure supplement 1B*) or with the annular lipids observed in the AQP01SM:2Chol structure, in which case the missing atoms in the lipid acyl chains were added (*Figure 5—figure supplement 1C*). For simulations using a tetramer pair, only the lipids sandwiched between them in the EC structures were included and the acyl chains completed as above (*Figure 5—figure supplement 1D*). Here, two different arrangements of sandwiched lipids were considered: either the layer consisting of the SM lipids seen in the AQP02SM:1Chol structure or a hybrid layer replacing the two central SM lipids with the EC deep cholesterol molecules observed in the AQP01SM:2Chol structure. In this case, both the tetramer pair and the sandwiched lipids were inserted in a membrane containing 1594 lipids.

Initial positions of the protein and annular lipids were taken from the EC structures of AQP0 with the surrounding annular lipids obtained in *N*-palmitoyl sphingomyelin (SM) membranes at low (~33%) and high (~66%) cholesterol (Chol) concentrations. Initial configurations of fully solvated membranes with three different cholesterol concentrations (see *Table 4*) were generated and equilibrated using the standard protocol of CHARMM-GUI (*Jo et al., 2008*; *Lee et al., 2016*). This equilibration process started with a Steepest Descent energy minimization step, followed by five consecutive short equilibration simulations (first three with a timestep of 1 fs, the final two with a timestep of 2 fs) with a gradual decrease in position and dihedral restraints for the lipid head groups and a final one without position restraints. This last step was performed for at least 100 ns until the bulk membrane properties, deuterium order parameter and area per lipid, became stable and congruent with previous reports such as those by *Doktorova et al., 2020*, and others (*Figure 5—figure supplement 2* and *Figure 5—figure supplement 3*). Large variations in the area per lipid occurred within the first few tens of nanoseconds, after which the area per lipid stabilized. In the particular case of the SM:Chol = 2:1 mixture, the 64 lipids/leaflet system converged to a stable area per lipid value in the last 70 ns and the 244 lipids/leaflet system approached the same value in approximately the last 30 ns. This was a good indication that the larger system had also converged.

AQP0 with or without the EC lipids was inserted into equilibrated SM:Chol membranes using g_membed (*Wolf et al., 2010*; *Yesylevskyy, 2007*). Five simulation replicas (10 for the tetramer pair) were run for each system with a simulation time per replica of 1 μs for the tetramer pair, and 500 ns for all others. For all simulations including EC lipids, the protein was inserted and then each replica was initialized with different velocity distributions. Upon insertion, the systems with one tetramer contained from 284 to 344 lipids and the systems containing the tetramer pair from 1292 to 1318 lipids. For simulations of a single AQP0 tetramer without EC lipids, in addition to using different random starting velocities, the protein was rotated at 18° intervals such that it faced different lipids. For simulations of a tetramer pair, the pair was rotated at 36° intervals for each replica. In the first five replicas, the tetramer pair was inserted at the center of the membrane, while in the other five replicas it was inserted at the corner of the pre-equilibrated lipid membrane. The first 100 ns of each simulation replica (the first 300 ns for the two tetramer simulations) were considered as additional equilibration time and were not included in further analysis.

All simulations were run using GROMACS (2019 version) (*Abraham et al., 2015*; *Páll et al., 2020*) using the CHARMM36 force field (*Brooks et al., 2009*). Periodic boundary conditions were applied. All systems were maintained at a temperature of 323 K (coupling constant of 1 ps) and a pressure of 1 bar (coupling constant of 5 ps) using the Nosé–Hoover thermostat (*Hoover, 1985*; *Nosé, 1984*) (the Berendsen thermostat (*Berendsen et al., 1984*) in the position-restrained equilibration steps) and the Parrinello–Rahman barostat (*Nosé and Klein, 1983*; *Parrinello and Rahman, 1981*) in a semi-isotropic scheme, respectively. The chosen temperature, which was also used in a previous MD study

of pure SM membranes (*Niemelä et al., 2004*), ensured that the lipid bilayers remained well above the liquid phase transition (*Keyvanloo et al., 2018*). For the Hydrogen atoms, we used the LINCS algorithm to solve the bond-length constraints (*Hess et al., 1997*), while the water molecules were constrained with the SETTLE algorithm (*Miyamoto and Kollman, 1992*). Simulations were run at a NaCl concentration of ~0.150 M and with the CHARMM TIP3P water model (*MacKerell et al., 1998*). Electrostatics were calculated using the particle mesh Ewald method (*Darden et al., 1993*; *Essmann et al., 1995*), splitting the direct and the reciprocal space at a distance of 0.12 nm. A Lennard–Jones potential was used to treat short-range interactions in a force-switch scheme transitioning from 1.0 nm until a cut-off of 1.2 nm. The Verlet buffer scheme was utilized to consider neighboring atoms (*Páll and Hess, 2013*).

### Force-probe MD simulations

Force-probe MD simulations on pairs of AQP0 tetramers were carried out using the same parameters used for the equilibrium simulations. Harmonic forces were applied to the center of mass of the backbone atoms of the α-helices of each tetramer, pulling them apart with a spring constant of 500 kJ/mol/nm$^2$ and moving the reference positions of the harmonic springs at a constant velocity at three different rates: 0.1 m/s, 0.02 m/s, and 0.004 m/s. 20 replicas were run for each rate, ranging from ~55 ns to ~900 ns. Initial configurations were taken from snapshots at 500 ns and 750 ns from each replica of the equilibrium simulations.

### Methods of analysis

#### Cholesterol insertion depth and orientation

The membrane-insertion depth and orientation angle of all cholesterol molecules of interest were measured using gmx trajectory and gmx gangle, respectively, from the GROMACS analysis toolbox (*Abraham et al., 2015*; *Páll et al., 2020*). The simulations of cholesterol in a pure SM bilayer, of an individual AQP0 tetramer with EC lipids, and of AQP0 tetramers pairs, were all considered independently for this calculation. In the case of pure SM bilayers, only the one inserted deep cholesterol molecule was analyzed, whereas for the simulations with an individual APQ0 tetramer, we monitored separately the closest cholesterol molecule to each S1 and S2 surface center (thus, the cholesterol at the deep position was dynamically selected and the EC cholesterol molecules were free to swap positions with the surrounding bulk ones). For simulations with pairs of AQP0 tetramers, due to symmetry, the two deep cholesterol molecules sandwiched between the tetramers were considered thus doubling the sampling.

The membrane-insertion depth was defined as the position of the cholesterol oxygen atom in reference to the average $z$ position of the phosphorus atoms of the SM molecules in the extracellular leaflet within 0.8 nm of the protein for the systems containing AQP0 tetramers, or within 3.0 nm of the monitored lipid for the systems not containing AQP0 tetramers ('no-AQP0'). Accordingly, the reference (i.e. $d=0$) was also dynamically set. The cholesterol angle was defined as the angle between the vector from the C-17 to the C-3 atom in the steroid ring and the $z$-axis (normal to the membrane plane).

#### Cholesterol distribution around AQP0 from unbiased simulations

The set of simulations with one AQP0 tetramer without the EC lipids was used to monitor the localization of cholesterol around an isolated AQP0 tetramer. The localization density was computed around every AQP0 monomer, which, due to the tetrameric structure of AQP0, effectively increased the sampling fourfold. Time-averaged density maps for the cholesterol localization were generated using GROma$\rho$s (*Briones et al., 2019*). In brief, a grid of 0.1 nm in resolution, spanning the simulation box, was considered. Atomic positions were spread onto the grid by a linear combination of four Gaussian curves with amplitudes and widths taken from *Hirai et al., 2007*. The density was computed separately around every monomer after least-square rigid-body fitting of their positions in the trajectories to the initial monomer position from the EC structure. The resulting four maps were then summed up using the GROma$\rho$s mapdiff tool. Density maps were visualized with PyMOL (The PyMOL Molecular Graphics System, Schrödinger, LLC) as isosurfaces at $9\sigma$ at a maximum distance of 1 nm away from the protein. The EC lipids for each lipid composition were overlaid to their corresponding SM:Chol ratio simulation.

## Principal component analysis

Principal component analysis (PCA) was carried out to analyze the relative motion of the two associated tetramers using the covar and anaeig tools from GROMACS. The covariance matrix of the positions of the backbone atoms was computed and diagonalized (*Amadei et al., 1993*). The equilibrium MD simulations with a pair of AQP0 tetramers containing an 'SM' interface were used for these calculations. Subsequently, both sets of trajectories (for the 'Chol' and 'SM' interfaces) were projected on the main principal components. A least-square fitting of the $C_\alpha$ atoms of one of the tetramers with respect to its initial conformation preceded the PCA calculation. Accordingly, the resulting trajectories correspond to the motion of one tetramer relative to the other (*Figure 7B*).

## Detachment force

The force acting on the tetramers and the separation between their centers of mass was monitored over time in the force-probe MD simulations (*Figure 7C and D*). The noise of these curves was reduced by applying a Gaussian filter to them (order 0 and Gaussian sigma width of 100 ps). The detachment force was assumed to be the maximum registered force before the separation of the centers of mass of the two tetramers surpassed 7.3 nm (*Figure 7D and E*). 20 replicas were performed for each system (*Figure 7E*). Box plots are presented as a function of the pulling velocity (*Figure 7F*). To estimate the significance and robustness of the difference in the detachment forces measured for the systems with and without deep cholesterol at the interface, we calculated the statistic and corresponding p-values, separately for each pulling velocity, using the Mann–Whitney U test (as implemented in SciPy 1.11.4, Python 3.9.19) (*Virtanen et al., 2020*), and considering the three pulling velocities at once, using the uncorrected two-way ANOVA tests (from the Statsmodel package, version 0.14.0; *Seabold and Perktold, 2010*).

## Molecular density maps between tetramers

The distribution of atoms at the interface between tetramers was assessed for the two paired AQP0 tetramers systems. This was achieved by computing the time-averaged density map on the surface of one tetramer that faced the adjacent tetramer. The equilibrium MD simulations of these systems were used as input for these computations. These maps were generated with GROma $\rho$ s (*Briones et al., 2019*) using the same calculation parameters as for the cholesterol density maps around an individual AQP0 tetramer (see above). To obtain the positional distribution of the different constituents of the system, the density was split into the contributions made by the adjacent tetramer, sphingomyelin, cholesterol, and water atoms.

## Protein–protein contacts

In the simulations of pairs of AQP0 tetramers, the total number of protein–protein contacts was computed using the GROMACS gmx mindist tool, using a cut-off distance of 0.6 nm to define a contact (*Figure 8—figure supplement 2*). In addition, the fraction of time one residue was in contact with a residue from the opposing tetramer, $T_{ij}$, was quantified using ConAn (*Mercadante et al., 2018*) and is shown in *Figure 8B*. The difference in this fraction for the pairs of AQP0 tetramers without (SM) or with deep cholesterols (Chol) at the interface, $\Delta T_{ij} = T_{ij}$(Chol)-$T_{ij}$(SM), was computed for each residue pair and is presented in *Figure 8C*. Finally, the average lifetime of each contact, $\tau_{ij}$, was also obtained with ConAn and is displayed in *Figure 8D*.

## Surface complementarity

For each of the two lipids at the center of the interface, sandwiched between AQP0 tetramers, we computed the contact area with each tetramer as $A_{Contact,i,m}=0.5(A_{Tet,m}+A_{lipid,i}-A_{Tet,m+Lipid,i})$. Here, $A_{Tet,m}$ is the surface area of tetramer $m$ ($m$=1,2), $A_{Lipid,i}$ is the surface area of the lipid $i$ (either SM or Chol, $i$=1,2) and $A_{Tet,m+Lipid,i}$ is the combined surface area of tetramer $m$ and lipid $i$. We then present $A_{Contact,i}=(A_{Contact,i,1}+A_{Contact,i,2})$. This value represents the amount of the surface of lipid $i$ that is covered by either tetramer, which, when normalized by $A_{Lipid,i}$, provides a measure of the surface complementarity between the lipids and tetramers. Consequently, a value of $A_{Contact}/A_{Lipid} = 1$ corresponds to a perfect match between the two molecular species (i.e. the entire surface area of the lipid is in contact with a tetramer), while $A_{Contact}/A_{Lipid} = 0$ indicates no match at all. In practice, neither of these two extremes

is observed and rather about half of the surface of a given lipid ($A_{Contact}/A_{Lipid}$ = 0.5) is in contact with one tetramer, and on fewer occasions partially with the two tetramers. $A_{Contact}/A_{Lipid}$ was computed separately for each of the two sandwiched central lipids located in the middle of the interface, either SM or Chol. Surface areas were computed using the double cubic lattice method, a variant of the Shrake–Rupley algorithm, as implemented in the gmx sasa GROMACS tool (*Abraham et al., 2015*; *Connolly, 1983*; *Eisenhaber et al., 1995*).

## Data availability

Model coordinates with the electron diffraction data in this study were deposited in the Protein Data Bank (PDB) under accession numbers 8SJY (AQP0$_{1SM:2Chol}$) and 8SJX (AQP0$_{2SM:1Chol}$). All data are available in the wwPDB database. MD simulations data have been deposited at the URL: https://github.com/MPTG-CBP/aqp0-chol.git, copy archived at *Chiu et al., 2024*.

## Acknowledgements

We thank Zongli Li and Andreas Schenk for support in data collection and processing and Chad Leidy and Juan Carlos Briceño for helpful discussions. This work was supported by National Institutes of Health grant R01 EY015107 (to TW), the Max Planck Tandem initiative at the Universidad de los Andes (to JDO and CAS), the Klaus Tschira Foundation (to CAS), and to BAYLAT (to JDO). We thank Frauke Gräter and Joachim Rädler for the PhD. student internship (to JDO). We acknowledge the Max Planck Computing & Data Facility in Garching, Germany, and the HPC data center of the Universidad de los Andes in Bogotá, Colombia, for computing time and computational resources.

## Additional information

### Funding

| Funder | Grant reference number | Author |
| --- | --- | --- |
| National Institutes of Health | EY015107 | Thomas Walz |
| Max Planck Tandem initiative at the Universidad de los Andes | | Juan D Orjuela<br>Camilo Aponte Santamaría |
| Klaus Tschira Stiftung | | Camilo Aponte Santamaría |
| Bayerisches Hochschulzentrum für Lateinamerika | | Juan D Orjuela |

The funders had no role in study design, data collection and interpretation, or the decision to submit the work for publication.

### Author contributions

Po-Lin Chiu, Juan D Orjuela, Data curation, Formal analysis, Investigation, Visualization, Writing – original draft, Writing – review and editing; Bert L de Groot, Supervision, Writing – review and editing; Camilo Aponte Santamaría, Thomas Walz, Conceptualization, Supervision, Funding acquisition, Writing – original draft, Writing – review and editing

### Author ORCIDs

Po-Lin Chiu ⓘ https://orcid.org/0000-0001-8608-7650
Bert L de Groot ⓘ https://orcid.org/0000-0003-3570-3534
Camilo Aponte Santamaría ⓘ https://orcid.org/0000-0002-8427-6965
Thomas Walz ⓘ https://orcid.org/0000-0003-2606-2835

Reviewer #1 (Public Review): https://doi.org/10.7554/eLife.90851.3.sa1
Reviewer #2 (Public Review): https://doi.org/10.7554/eLife.90851.3.sa2
Reviewer #3 (Public Review): https://doi.org/10.7554/eLife.90851.3.sa3

Author response https://doi.org/10.7554/eLife.90851.3.sa4

## Additional files

### Supplementary files
• MDAR checklist

### Data availability
Electron diffraction data have been deposited in PDB under the accession codes 8SJY and 8SJX. Molecular dynamics simulations data have been deposited on GitHub (copy archived at *Chiu et al., 2024*).

The following datasets were generated:

| Author(s) | Year | Dataset title | Dataset URL | Database and Identifier |
|---|---|---|---|---|
| Chiu PL, Walz T | 2024 | Structure of lens aquaporin-0 array in sphingomyelin/cholesterol bilayer (1SM:2Chol) | https://www.rcsb.org/structure/8SJY | Protein Data Bank, 8SJY |
| Chiu PL, Walz T | 2024 | Structure of lens aquaporin-0 array in sphingomyelin/cholesterol bilayer (2SM:1Chol) | https://www.rcsb.org/structure/8SJX | Protein Data Bank, 8SJX |

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
