## [Editor Report · eLife assessment]

This manuscript aims to unravel the contribution of cholesterol to aquaporin-0 (AQP0) tetramer array formation within lens membranes. **Compelling** electron crystallography data are combined with **solid** molecular dynamics experiments to identify a specific cholesterol binding site of significance to protein clustering within lipid rafts. The **important** work advances our understanding of membrane biology and will be of broad interest to membrane transport biologists, biochemists, and structural biologists.

---

## [Referee Report · Reviewer #1 (Public Review)]

Aquaporin-0 forms 2D crystals in the lens of the eye. This propensity to form 2D crystals was originally exploited to solve the structure of aquaporin-0 reconstituted in membranes. Existing structures do not explain why the proteins spontaneously form these arrays, however. In this work the authors investigate the hypothesis that the main lipids in the native membranes, sphingomyelin and cholesterol, contribute to lattice formation. By titrating the cholesterol: sphingomyelin ratio, the authors identify cholesterol binding sites of increasing stability. The authors identify a cholesterol that interacts with adjacent tetramers and is bound at an unusual membrane depth. Computational simulations suggest that this cholesterol is only stable in the context of adjacent tetramers (ie lattice formation) and that the presence of the cholesterol increases the stability of that interface. The exact mechanism is not clear, but the authors propose that the so-called "deep cholesterol" improves shape complementarity between adjacent tetramers and modulates the kinetics of protein-protein interactions. Finally, the authors provide a reasonable model for the role of cholesterol in

Strengths of this manuscript include the analysis of multiple structures determined with different lipid compositions and lipid:cholesterol ratios. For each of these, multiple lipids can be modelled, giving a good sense of the lipid specificity at various favorable lipid binding positions. In addition, multiple hypotheses are tested in a very thorough computational analysis that provides the framework for interpreting the structural observations. The authors also provide a thorough scholarly discussion that connects their work with other studies of membrane protein-cholesterol interactions.

The model presented by the authors is consistent with the data described.

---

## [Referee Report · Reviewer #2 (Public Review)]

Summary:

In the manuscript by Chiu et al., "Structure and dynamics of cholesterol-mediated aquaporin-0 arrays and implications for lipid rafts," the authors address the effect of cholesterol on array formation by AQP0. Using a combination of electron crystallography and molecular dynamics simulations, the authors show binding of a "deep" cholesterol molecule between AQP0 tetramers. Each AQP0 tetramer binds four deep cholesterols to form a crystallographic array of AQP0.

Strengths:

The combined approaches of electron crystallography and MD simulations under different lipid conditions (different sphingomyelin and cholesterol concentrations) are a strength of the study. The authors provide a thorough and convincing assessment of cholesterol binding, protein-protein interactions, and array formation by AQP0. The MD simulations allow the authors to consider the propensity of cholesterol to occupy the observed binding sites in the absence of crystal contacts. The combined methods and the breadth of analyses set a high standard in the field of membrane protein structural biology.

The findings of the authors fit nicely into a growing body of literature on cholesterol binding sites that mediate membrane protein-protein interactions. Cholesterol interacts with a variety of membrane proteins via its smooth alpha face of rough beta face. AQP0 is somewhat unique in that it binds the rough face of cholesterol in a "deep" binding site that places cholesterol in the middle of the membrane bilayer. So-called "deep" cholesterol binding sites have been described for GPCRs and docking studies suggest they may exist on other ion channels and transporters. In the case of AQP0, the deep cholesterol acts as a glue that holds two tetramers together. Since each tetramer has four binding sites for deep cholesterol, the assembly and mechanical stability of an extended two-dimensional array of AQP0 tetramers is a natural consequence in lens membranes.

Weaknesses:

The authors report that the findings generally apply to raft formation in membranes. However, this point is less clear as the lens membrane in which AQP0 resides is rather unique in lipid and protein content and density. Nonetheless, the authors achieve the overall goal of evaluating cholesterol binding to AQP0, and there are many valuable and informative figures in the main manuscript and supplement that provide convincing results and interpretations.

---

## [Referee Report · Reviewer #3 (Public Review)]

Summary:

This manuscript aims to unravel the mechanisms behind Aquaporin-0 (AQP0) tetramer array formation within lens membranes. The authors utilized electron crystallography and molecular dynamics (MD) simulations to shed light on the role of cholesterol in shaping the structural organization of AQP0. The evidence suggests that cholesterol not only defines the positions and orientations of associated molecules but also plays a crucial role in stabilizing AQP0 tetramer arrays. This study provides valuable insights into the potential principles driving protein clustering within lipid rafts, advancing our understanding of membrane biology.

In this review, I will focus on the MD simulations part, since this is my area of expertise. The authors conducted an impressive set of MD simulations aiming at understanding the role of cholesterol in structural organization of AQP0 arrays. These simulations clearly demonstrate the well-defined localization of cholesterol molecules around a single AQP0 tetramer, aligning with previous computational studies and the crystallographic structures presented in this manuscript. Interestingly, the authors identified an unusual position for one cholesterol molecule, located near the center of the lipid bilayer, which was stabilized by the adjacent AQP0 tetramers. The authors showed that these adjacent tetramers can withstand a larger lateral detachment force when deep cholesterol molecules are present at the interface compared to scenarios with sphingomyelin (SM) molecules at the interface between two AQP0 tetramers. Authors interpret that result as evidence that deep cholesterol molecules mechanically stabilize the interface of the AQP0 tetramers.

The simple steered MD simulations are typically employed to either identify pathways for subsequent free energy calculations, such as umbrella sampling or perform numerous non-equilibrium simulations, utilizing the Jarzynski equation to extract free energy. In this paper, the authors conducted steered MD simulations to examine the maximum force required to separate tetramers, and they did not carry out the more rigorous but challenging free energy calculations. The observation that the maximum force needed to separate tetramers in the presence of cholesterol (compared to the SM case) suggests a positive direction in the authors' work, however, free energy calculations would be needed to fully support the cholesterol stabilization effect.

---

## [Author Response]

The following is the authors’ response to the previous reviews.

**Reviewer #1 (Public Review):**
The model presented by the authors is consistent with the data described. Further testing of this model, for example by mutating the deep cholesterol binding site, would strengthen the model. However, such experiments might be challenging due to the relatively non-specific/hydrophobic nature of the deep cholesterol binding site.

We completely agree that testing of the deep cholesterol-binding site by mutagenesis would be ideal. However, as the reviewer points out, such experiments would be challenging, not only because of the non-specific/hydrophobic nature of the deep cholesterol-binding site but also because we have been purifying AQP0 from natural sources (sheep eyes) and because it would be very difficult to secure the substantial amount of cryo-EM time needed to generate an electron crystallographic structure.

**Reviewer #2 (Public Review):**
The authors report that the findings generally apply to raft formation in membranes. However, this point is less clear as the lens membrane in which AQP0 resides is rather unique in lipid and protein content and density.

We agree that the lens membrane is quite unique in its lipid and protein content and density, but rafts are also characterized by the same lipids and high protein density. Nonetheless, we do agree that our suggested implications for lipid rafts are speculative and so we emphasize this more in the revised version of the manuscript by writing: “This model is specific for the formation of AQP0 arrays in lens membranes, but we speculate that similar principles may underlie the organization of lipid rafts”.

**Reviewer #3 (Public Review):**
The authors showed that these adjacent tetramers can withstand a larger lateral detachment force when deep cholesterol molecules are present at the interface compared to scenarios with sphingomyelin (SM) molecules at the interface between two AQP0 tetramers. Authors interpret that result as evidence that deep cholesterol molecules mechanically stabilize the interface of the AQP0 tetramers. This conclusion has minor weaknesses, and the rigor of the lateral detachment simulations could be increased by establishing a reference point for the detachment force needed to separate AQP0 tetramers in a scenario without lipids at the interface between tetramers, and by increasing the number of repeats for the non-equilibrium steered MD simulations. Thermodynamic integration might be a better approach to compute the stabilization energy in the presence of cholesterol compared to the SM case.

In all electron crystallographic structures of AQP0 determined to date, lipids have always been observed sandwiched in between the AQP0 tetramers (see, for example, Gonen et al., Nature, 2005 and Hite et al., EMBO J., 2010). Therefore, considering a scenario without lipids at the interface would be unnatural and the AQP0 array would likely not be stable. Such a scenario would thus not be the most appropriate reference point for the lateral detachment simulations. In our view, comparison of a scenario with the deep cholesterol at the interface *versus* a scenario without it appeared a more realistic setup to investigate the stabilizing role the deep cholesterol has on the association of AQP0 tetramers. In the Results subsection regarding these simulations, we added the following sentence to further stress the rationale of our experimental setup: “Comparison of these two cases should allow us to assess the effect of the deep-binding Chol3 molecules on the mechanical stability of the associated AQP0 tetramers.”

Concerning the second suggestion of the reviewer of increasing the number of repeats, we doubled the number of simulation replicas: now it is n=20 for each pulling velocity and lipid interface. The trend of higher detachment forces for the interface containing cholesterol prevailed in a statistically significant, robust fashion (see Figure 7 of the revised manuscript and the main text referring to it). In consequence, as the reviewer suggested, extension of the dataset increased the rigor of the lateral detachment simulations. In addition to Figure 7 and the Results section, the Methods section and Table 4 have been updated to reflect the expanded dataset.

Finally, concerning the usage of thermodynamic integration to compute the stabilization energy, we agree with the reviewer that calculation of the free energy would be better to determine the thermodynamic stabilization imparted by the cholesterol molecules. At an earlier stage of the project, we did indeed consider carrying out this type of simulations, but we decided against it because of the complexity and poor convergence of such calculations. Our choice is also based on a previous attempt in which it proved very challenging to use free energy calculations to assess the binding of lipids to a flippase (see [120]). We now included this consideration in the revised manuscript by adding the following sentence in the Discussion: “Although we provide solid evidence here that deep cholesterol impart mechanical stabilization, free energy calculations would be required to obtain the full picture of thermodynamic stabilization. Such free energy calculations are challenging for lipids, due to the chemical complexity and poor convergence involved (120), and are thus beyond the scope of the current work.”

**Reviewer #1 (Recommendations For The Authors):**
Reorganizing a few concepts would make the story easier to follow. For example, the analysis of the bilayer thickness seems disjointed. Although Figure 4 shows measurements, it is not clear that the measurements represent bilayer thickness until the last paragraph of page 21 in the discussion, where "Hydrophobic thickness" is first introduced. Moving that first paragraph of page 22 that refers to Fig. 4A to the results would be helpful to understand the figure, and would prepare the reader for this part of the discussion.

In response to the reviewer, we moved the description of the measurements of the hydrophobic thickness to the Results section (Page 12) and adjusted the Discussion to minimize repetition (page 22).

Likewise, Figure 4E shows measurements of something, but it is not clear that these are the dimensions of a protein pocket until well into the discussion.

In response to the reviewer’s comment, we added a sentence both in the Results section [It sits in a pocket between the two adjacent AQP0 tetramers that is wider in the extracellular leaflet than the cytoplasmic leaflet (Figure 4E)] as well as to the caption of Figure 4E [The dotted lines indicate the distance between the two adjacent AQP0 tetramers at the positions of the ring system (~8.5 Å) and the acyl chain (~2.5 Å)].

Figure 2 - a comment for the non-specialists on what this region of the protein is would be helpful context. Is this the pore with part of the NPA motif?

We agree with the referee and added the following sentence to the caption of Figure 2: “A region of the water-conducting pathway close to the NPA (asparagine-proline-alanine), the AQP signature motif, is shown”.

**Reviewer #2 (Recommendations For The Authors):**
There is only one recommendation: In the results section entitled "Cholesterol positions observed in the electron crystallographic structures are representative of those around single AQP0 tetramers" the authors do not describe their approach. They refer to a reference (AponteSantamaria et al., 2012). The authors state the problem (investigate cholesterol positions), but it would be helpful to the readers if they also described the experimental approach.

We agree with the reviewer and made the following addition to the sentence “we performed MD simulations and calculated time-averaged densities to investigate ...”

**Reviewer #3 (Recommendations For The Authors):**
Technical comments:(1) Authors stated: "Equilibration simulations were then performed until bulk membrane properties, such as thickness and deuterium order parameters, became stable and congruent with previous reports such as those by (Doktorova et al., 2020) and others (Figure 5-figure supplement 2 and Figure 5-figure supplement 3)." However, bilayer thickness is not represented in these figures. Additionally, I observed that the area per lipid (APL) appeared to be somewhat variable. This variation was particularly noticeable in systems where SM:CHOL=2:1, which seem to be not fully equilibrated. Is the figure displaying APL data for only one repetition? Could you please include plots for the other repetitions?

We thank the reviewer for pointing this out. We would like to clarify that we used CHARMMGUI to generate one lipid bilayer configuration for each mixture and system size. These configurations (one per system) were extensively simulated to generate stable initial configurations of the lipid bilayers. Figure 5 – supplements 2 and 3 refer to this pre-equilibration step. The final pre-equilibrated configurations were then used in the subsequent multiple equilibrium MD runs that we performed, either with a single cholesterol molecule or with the AQP0 tetramer(s) inserted. We have clarified this procedure in the revised manuscript (see changes in the Methods section for the MD equilibrium simulations).

Concerning this pre-equilibration step, we have chosen the area per lipid, not thickness, to characterize the equilibration of the pure lipid bilayers. Accordingly, the area per lipid is the quantity shown in Figure 5 – figure supplement 3. We no longer refer to the membrane thickness in the revised manuscript.

Concerning the variability in the area per lipid, we note that the large changes occur within the first few tens of nanoseconds of the pre-equilibration step, after which the area per lipid stabilizes. We would like to also point out that in Figure 5 – figure supplement 3, we chose a logarithmic scale for the time axis to actually make it possible for the reader to see the major changes that occur at the beginning of the pre-equilibration step (which would otherwise be difficult to see). In the particular case of the SM:CHOL=2:1 mixture_,_ the 64 lipids/leaflet system converged to a stable area per lipid value in the last 70 ns and the 244 lipids/leaflet system approached the same value in approximately the last 30 ns. This was a good indication that the large system had also converged. After equilibration of the membranes, a single cholesterol or AQP0 tetramer(s) were inserted and equilibrium simulations were initiated. However, the first 100 ns (or 300 ns in the case of the double tetramer system) were considered as a further equilibration time and were not included in the analysis. This is now explicitly stated in the revised manuscript: “The first 100 ns of each simulation replica (the first 300 ns for the two tetramer simulations) were considered as additional equilibration time and were not included in further analysis.”

(2) Could you clarify the reasoning behind conducting the simulations at 323 K?

We conducted the simulations at 323 K to ensure that the lipid bilayers were in the liquid phase.

SM:CHOL mixtures have been reported to be in the liquid phase above 314 K (Keyvanloo et al. Biophys. J. 114: 1344, 2018). 323 K was thus chosen to be well above this value. Note that this temperature was also chosen in a previous MD simulation study of pure sphyngomyelin bilayers (Niemelä et al. Biophys. J. 87: 2976, 2004). This reasoning, as well as the two references, have been added to the Methods section in the revised manuscript.

(3) There appears to be a discrepancy in Figure 7. Panel F does not align with the provided caption.

We apologize for this mistake. The captions for panels E and F were switched. We corrected this mistake.

(4) Likewise, in Figure 8, there is a mismatch between the caption and the figures. Furthermore, in the text, the authors assert, "In the absence of cholesterol, the AQP0 surface is completely covered by sphingomyelin in the hydrophobic region of the membrane and by water outside this region (Figure 8A, left column). As noted before, there are essentially no direct protein-protein interactions between the adjacent tetramers. When cholesterol was present at the interface, it interacted with AQP0 at the center of the membrane and remained mostly in place (Figure 8A, right column)." However, the left column shows cholesterol density. Could you please clarify this inconsistency, especially regarding the absence of cholesterol?

We apologize for this mistake. The panels in Figure 8A showing the AQP0 surfaces in the absence and presence of cholesterol were switched. We corrected this mistake.